# Computer Vision Algorithms of DigitSeis for Building a Vectorised Dataset of Historical Seismograms from the Archive of Royal Observatory of Belgium

**DOI:** 10.3390/s23010056

**Published:** 2022-12-21

**Authors:** Polina Lemenkova, Raphaël De Plaen, Thomas Lecocq, Olivier Debeir

**Affiliations:** 1Laboratory of Image Synthesis and Analysis (LISA), École Polytechnique de Bruxelles (Brussels Faculty of Engineering), Université Libre de Bruxelles (ULB), Building L, Campus de Solbosch, Avenue Franklin Roosevelt 50, BE-1050 Brussels, Belgium; 2Royal Observatory of Belgium, Seismology & Gravimetry (OD2), Avenue Circulaire 3, BE-1180 Uccle, Belgium

**Keywords:** seismology, Galitzine seismometer, horizontal component, analogue seismogram, digitising, earthquake recording, ground motions, historical seismograms, seismic waves

## Abstract

Archived seismograms recorded in the 20th century present a valuable source of information for monitoring earthquake activity. However, old data, which are only available as scanned paper-based images should be digitised and converted from raster to vector format prior to reuse for geophysical modelling. Seismograms have special characteristics and specific featuresrecorded by a seismometer and encrypted in the images: signal trace lines, minute time gaps, timing and wave amplitudes. This information should be recognised and interpreted automatically when processing archives of seismograms containing large collections of data. The objective was to automatically digitise historical seismograms obtained from the archives of the Royal Observatory of Belgium (ROB). The images were originallyrecorded by the Galitzine seismometer in 1954 in Uccle seismic station, Belgium. A dataset included 145 TIFF images which required automatic approach of data processing. Software for digitising seismograms are limited and many have disadvantages. We applied the DigitSeis for machine-based vectorisation and reported here a full workflowof data processing. This included pattern recognition, classification, digitising, corrections and converting TIFFs to the digital vector format. The generated contours of signals were presented as time series and converted into digital format (mat files) which indicated information on ground motion signals contained in analog seismograms. We performed the quality control of the digitised traces in Python to evaluate the discriminating functionality of seismic signals by DigitSeis. We shown a robust approach of DigitSeis as a powerful toolset for processing analog seismic signals. The graphical visualisation of signal traces and analysis of the performed vectorisation results shown that the algorithms of data processing performed accurately and can be recommended in similar applications of seismic signal processing in future related works in geophysical research.

## 1. Introduction

### 1.1. Background

The seismicity of the Earth represents a physical phenomenon resulting from the tectonic processes of energy accumulation and release [1]. This fundamental concept of the Earth’s physics is reflected in the movements on the surface that have a different intensity of the vibrations caused by the fluctuations of energy [2]. In geodynamics, the seismicity is the result of the self-organisation of the Earth which responds to the lithosphere movements [3]. Measuring seismic signals using seismometers enables to evaluate ground motions of the Earth associated with the earthquakes of various magnitude or ambient seismic noise [4]. Evaluating seismic ambient noise is effective in different contexts, including imaging and monitoring the Earth’s interior, seismic mapping at a global or regional scale, environmental studies, investigations on elastic deformations at shallow depths in the crust, to mention a few.

The EO data processing in seismology has undergone a long process of technical evolution [5,6]. The instruments used for seismic data recording went through a constant development of technologies and state-of-the-art methods. The systematic, precise and operative observation of the seismicity of the Earth began in late 19th century along with a progress in technology, development of new seismographs and advances in geophysical methods [7]. From the 1880s to the 1960s, seismographs were used for recording ground motion continuously and systematically [7]. The data were stored in an analogue paper-based format using narrow-band low-range instruments [8,9] or as magnetic tapes [10].

Starting in the 1960s, seismology benefited from the onset of the computer era that led to a transition to digital data which increased in quantity worldwide along with instrumental development [11]. A global initiative of seismological network WWSSN was created in 1960s to generate a collection of the high-quality big seismic datasets [12,13]. This initiative was possible due to the uniformly accepted technical equipment and standardised workflow for data calibration in each seismic station. Currently, seismograms are recorded and stored digitally in high-performance data centres using advanced algorithms of signal processing [14]. However, the historical datasets from the pre-digital period remain a valuable database containing unique information on seismicity of the Earth that is useful for practical scientific purposes [15,16,17,18,19].

Processing these datasets is important for practical applications in the fields of seismology, geology, tectonics and physics of the Earth. The wide use of information retrieved from the seismograms [20] includes such domains as geophysics [21], ocean climate modelling [22], subsurface exploration, global and planetary tectonics [23,24,25], risk assessment in civil engineering [26,27,28], earthquakerisk analysis with associated features (dynamic rupture, seismic tomography and wave forms) [29,30,31,32,33], and volcanology [34].

Recently, with the advances in studies of the ambient wavefield, or “noise”-based techniques, it has become possible to use the data recorded between the earthquakes to conduct monitoring [35,36,37]. Moreover, the advanced methods of data processing facilitate signal and image analysis in geoscience and geophysics in various contexts [38,39,40,41,42,43]. Aside from the terrestrial regions, oceans are another important global source of seismic energy [44,45,46]. They mostly generate surface waves that travel large distances and can affect remotely located areas. The microseisms are the dominant component of the ground motions recorded at the Earth’s surface. In turn, the analysis of microseisms allows deeper insights into the evolution of oceanic climate [47,48] and the dynamics of the atmosphere–ocean–earth couplings [44,49,50].

### 1.2. Motivation

Old analog seismograms present a key source of information regarding the seismicity of the Earth in the pre-digital age. There is also a time-sensitive component of digitising seismograms before they are degraded and lost over time. The importance of the historical datasets for geophysical research raise the need to ensure their digital conservation [51,52,53,54]. Using archived information, it is also possible to analyse past seismicity in historical contexts [55,56,57], which is important for geophysical reconstructions and climate modelling. Datasets on seismic velocity as time series can be used for monitoring volcanoes. Moreover, retrospective analysis of the old seismograms enables to model earthquake cycles as a continuous process and to gain deeper insights into the physics of the Earth. Finally, the interpretation of the old seismograms provides an important information for the long-term prognosis. For instance, datasets on earthquakes, including locations and magnitude, are essential for predictive modelling [58].

Such analyses are largely based on the retrieval of information from the digital seismograms [59]. However, the use of such information relies heavily on our access to large historical datasets of seismograms. While paper-based seismograms archived in observatories contain valuable scientific information, they are not suitable to reuse in data modelling. In contrast, seismograms in vector format present uniformed and standardised data collections in digital format that can be accurately processed. Processing such datasets enables to explain the physical nature of the ground motion signals from the Earth’s interior. The updates of such datasets depends on the accurate and standardised recording systems using commonly accepted and approved instruments (seismometers and seismographs) with products that remain consistent through the decades of operation [3].

### 1.3. Related Work

The first attempts to digitise the analogue recordings of seismic events and convert the data into numeric format started in 1960s. An ad hoc algorithm of thinning the lines on scanned seismograms and converting the image into a 0–1 digital matrix was developed by [60]. Another study used a device initially designed for weather maps and adjusted for automatic converting of seismic signals into a digital form and plotting them using a cathode-ray tube display [61]. The progress of digitising signals accelerated in 1980s [62,63] and 1990s [64,65,66,67,68,69] along with the increase of the computer-based algorithms and developed geophysical software. The digitising approaches have become more and more automated since 2000s, along with a rapid development of the programming languages and IT tools [9,70,71,72].

Recently, the use of the automated digitisers of seismograms instead of the manual vectorisers has received much attention. They enable vectorising the recordings of the acceleration of the strong motion ground. The advanced algorithms include the automated recognition of noise [63], handling smooth and wigged traces for tracing waves [70], synchronisation of the time scale for the three motion components, handling scratches, distortions or line crossings on the image, and adjusting rotated position [73].

Digitising old seismograms is a not straightforward task. Often the problems are caused by the recording methods in old mechanical seismographs. For instance, high velocity of stylus that does not touch the paper, clipping of amplitudes, low-contrast dark background on smoked paper, records damaged over time due to bad storage conditions of paper [74], to mention some of them. At the same time, the digitising process should be as automated as possible, to avoid human-induced errors. As a response to these needs, various programs for automatic, quick and accurate digitising and modelling of seismograms were developed. The advances in pattern recognition are supported by rapid growth of programming languages [75,76,77,78,79], and specifically, Python [80,81,82], improvements of algorithms of ML for image segmentation and clustering [83,84,85,86,87], and signal processing [88,89].

Pattern recognition is a key basis for digitising seismograms using in many existing software. For example, the Teseo [90] traces raster files using cubic Bézier curves handling monochrome images by a combination of manual and automatic methods and neural networks. A software using C# uses an algorithm of colour scene recognition by Color Scene Filed Method [91]. A MATLAB-based program used algorithm of inversion problem for a matrix of model parameters and observed seismic data [92]. Another example of seismic software, the SeisDig, developed by Berkeley Seismological Laboratory, presents an interactive MATLAB-based tool which performs trace inspection, checks time conservation, recognises events and inputs metadata header [93]. An approach to seismic recording is presented in DigiSeis [94] aimed at digitizing signals using a built-in PC sound card for signal processing. It visualises the two-channel seismic data in time and frequency, and performs time series analysis. Extracting wave data from paper seismograms was attempted by [95] with a seismogram digitization and DB management system using Delphi3 approach based on Pascal.

### 1.4. Contribution

As a response to the needs for accurate vectorising of old archived seismograms, the goal of our study was to perform a digitisation of seismic traces contained in historical seismograms. To this end, we applied the DigitSeis software [96] for semi-automated vectorising of seismograms. The aim was to convert an existing large historical collection of seismograms archived in Royal Observatory of Belgium (ROB) from raster format into a vector format using advanced technical functionality of machine processing in an automatic and accurate way. The dataset included over 140 scanned raster images in TIFF format which were converted into the digital MATLAB output with generated .mat format files.

The objective was to update the historical dataset of the old seismograms recorded at Uccle station in 1954 by the Galitzine seismometer. We scanned the large volume of archived seismogram data and vectorised the files using DigitSeis. Our study contributes to the maintaining methods of vectorising archived seismograms for information retrieval and analysis. We also identified the limits and possibilities of the vectorisation tools to propose a framework of using machine methods for geophysics and seismology.

The proposed framework benefits from simple, efficient pixelwise processing of analogue paper-based seismograms, which is easily amenable to various data formats. Moreover, the use of DigitSeis does not require the pre-processing of time gaps and marks, initialization strategies, neither has smoothness constraints with regard to image resolution. Other advantages include the adaptability to structure of recorded papers and easy recognition of seismic traces, including fine details. Consequently, our approach is not biased by instability for the dataset containing seismograms recorded by the same instrument (Galitzine), and we do not have difficulty handling materials with varying records, e.g., records containing noise or high-frequency waves. Extensive experiments on vectorised seismograms shown that our implementation using robust algorithms of DigitSeis produces stable, accurate, and efficient vectorisation of analogue seismograms.

## 2. Materials and Methods

### 2.1. Study Area

The study was performed in the Université Libre de Bruxelles, École Polytechnique de Bruxelles, Laboratory of Image Synthesis and Analysis (LISA), Belgium. We used archives obtained as a courtesy of ROB, Department of Seismology & Gravimetry, UCC, Figure 1. The data processing is performed within the framework of the SeismoStorm project.

Established in 1899 by Eugène Lagrange, Uccle was the only seismic station in Belgium for the most of the 20th century. Afterwards seismic monitoring gradually expanded to a larger seismic network system of Belgium [98]. Nowadays, the Royal Observatory of Belgium (ROB) located in Uccle remains an important centre of Belgian seismic network. It collects seismic data from 28 measuring stations of Belgium and Luxembourg, and maintains the archiving of the seismogram database.

### 2.2. Instrument

The dataset obtained from the Galitzine seismometer was processed using the DigitSeis vectoriser. The original seismograms have been received from UCC using the Galitzine seismometer, Figure 2. It is the 1st electromagnetic instrument developed by B.B. Galitzine in 1910 (Figure 2) [99,100], designed to record a single component (horizontal or vertical) for earthquake recording [101]. This type of seismographs is constructed using a following principle: a coil of wire is fixed to the pendulum which oscillates between the poles driven by the earthquake forces. The movement of coil between a strong magnet generates an electric induction current. The current is being transferred to a sensitive galvanometer to direct a beam of light onto the photographic paper which visualizes the strength of the signals [102].

The structure of the Galitzine seismometer is presented in Figure 2. The pendulum (at the front), consists of a mass suspended by the two wires. It includes several parts: copper plate fixed on the rod next to the coil; electromagnetic seismometer with a coil attached to the arm of the pendulum which oscillates in a magnetic field between the poles of a magnet (at the back) and generates an electric current by induction; Foucault current for damping; horizontal component (7 kg); tracing device with an optical recording system which includes a photographic paper and a galvanometer mirror with a mobile frame; and the speed settings with a period of 12 s.

B.B. Galitzine improved the accuracy of the seismometer by reducing the sensitivity of the system through diverting a part of the current. Thus, the Galitzine seismometer includes the two special features: (1) Damping the resistance to the galvanometer; (2) Damping the seismometer pendulum through the cumulated effect of a magnetic damping device and the external resistance to the pendulum coil. Due to such effectiveness, this principle is widely used for seismic measurements by the Galitzine-type seismometers [103].

### 2.3. Workflow

The general flowchart scheme for data processing is shown in Figure 3. This study is based on integrating the approach which combines several methods. The data were collected from the archives of ROB as historical seismograms recorded by the Galitzine seismometer in 1954. In 2021, the data were scanned in A0 format in a very-high resolution (600 dpi) and saved as TIFFs which were imported to DigitSeis and converted into 8-bit native HSV format.

The next steps included stepwise data processing in DigitSeis. The pre-processing included image loading, measuring time gaps and navigating selected segments. The object classification included the setup of image threshold, marking time parameters and exploring small regions for detailed trace analysis. Major steps included vectorisation, digitising. Finally, the post-processing of data consisted in converting the output files into the .mat format. Most of the traces were well recognised by the algorithms of DigitSeis during digitising, while selected segments required interactive corrections which were repeated iteratively for each seismogram. The results of DigitSeis data processing shown a relatively good recognition of lines and traces. The validation of data was performed for the exported files.

The digitised seismic traces present a series of traces with time gaps identified semi-automatically. The files were converted into .mat format and processed as the next steps using Python. The Matplotlib library was used for post-processing steps and quality control. We analysed the repeatability of the segments broken by DigitSeis and visualised the received vector traces as separate plots. We checked the time ticks which were automatically recognised by the program per minute using time gaps, and segments of the lines.

### 2.4. Data

The dataset received from the Uccle station presents a large collection of scanned paper-based seismograms with selected examples presented in Figure 4.

The images are stored as TIFF files which present a matrix of 2D arrays of pixels. Each pixel is represented by a corresponding number of grayscale [104]. The images were stored at a resolution of 300 dpi, ca. 350–400 MB size each, in 256 B/W monochrome scales. The standard format of all the images ensured the integrity and compatibility of the whole dataset used for data analysis. The resolution of 300 dpi was considered enough for the ambient seismic noise analysis, even though some files were originally saved at 1200 dpi. The records from the Galitzine seismometer are monochrome while those of Wiecherts are stored in color.

The dataset included a collection of 145 images covering the period from 1 January 1954 to 12 March 1954. Some of the images are well preserved, while others have distortions and defects visible on the aged paper, Figure 4. The scanned images of old seismograms have different features and characteristics, which increase technical difficulties of processing such data: line breaks, distortions, defects, blurred lines, spots, etc. For instance, some data had defects due to the record imperfectness or various cases related to storage. Oftentimes, the traces are slanted as shown, e.g., in Figure 6, while others appear more horizontal, see Figure 7. This is basically caused by the physical features of spiral recording by a rotating drum of the seismometer. Technical tasks concerned the quality of the recordings on such data with each individual seismogram and challenges to the semi- automatic recognition of traces on the old scanned paper.

### 2.5. Software and Workflow

The software used in this study included DigitSeis, a MATLAB based software for processing analog scanned seismograms [96]. We followed a structured workflow for data processing, Figure 5.

We used the DigitSeis software version 1.5, developed by Bogiatzis, P. and Ishii, M. from the Department of Earth and Planetary Sciences, Harvard University, Cambridge, MA, USA. The DigitSeis is developed as an instrument for digitising scanned seismograms. It extracts the data from the seismograms and processes them as an image matrix. The DigitSeis converts scanned raster images of analog seismograms from the original TIFF format into the convenient digital format using algorithms of signal processing. The approach of the DigitSeis is based on the principles of semi-automated algorithms of signal processing as time series. It applies minor human supervision, mostly for the adjustment of the functionality in selected process steps and for monitoring technical workflow (Figure 6).

The workflow of seismogram vectorisation by DigitSeis can be summarised in five major steps as follows: (1) Image preprocessing; (2) Identification of time marks and traces; (3) Image vectorising and classification; (4) Trace correction and control; (5) Timing and conversion (SAC).

An algorithm embedded in DigitSeis uses trace analyses which enables to track the lines and to record their values using time gaps. We applied this method to identify traces on the scanned TIFF files of seismograms and processed them by a semi-automatic detection of lines and distortions. The latter include, for instance, broken lines, spots, repetitive gaps on the series of lines. The digitising was adjusted with aim to distinguish the skeleton of lines, similar to the ‘0–1’ matrix, using methods of colour recognition on the image. However, the directions of vector lines present another task that can only be solved using the advanced analysis of the geometric trend of the trace by the automated tools. The thickness of the traces and the time gaps were considered for each 30-min line on each seismogram.

#### 2.5.1. Image Preprocessing

First, the images were loaded into the program as TIFF files and converted into the native 8-bit HSV formal of DigitSeis. The intensity of the monochrome images is either 0 (black) or 255 (white) in terms of grayscale. Visually, the image contained white traces of the measured seismic signals over a black background (Figure 6). In the selected images with limited data and large dark background, the areas of interest were cropped in order to reduce the space without traces and to select only a part of image to test a smaller space area of interest (Figure 7). The resulted analysis was saved continuously throughout the workflow into a in the binary data MATLAB format as a .mat file.

#### 2.5.2. Identifying theTime Gaps

The interpretation of seismic reflection signals involves the identification of the arrival times recorded in the time markers or gaps. This is important step, because the exact determination of time and coordinates of the signals is essential in seismology.

Therefore, we evaluated the 1-s time mark dimensions which show the breaks of the line along its course. The recording contained the time marks at every minute. The whole line represents, on average, 30 min intervals of the ground motion signals (Figure 7). On the original TIFFs of the seismograms, the time is marked according to the UTC time and most of the times is recorded at 30 mm/min. Marking time was performed by measuring width and the offset of gaps in the traces and recording the values in a menu bar. The time intervals on the seismograms can be represented in two ways: time marks (small vertical dashed lines) or time gaps (small breaks between the lines). In this study, we only had time gaps due to the technical specifics of the Galitzine seismometer. In DigitSeis, the time marks are indicated as positive numbers, while time gaps—as negative ones. Therefore, in our case the time intervals were indicated as negative numbers, Figure 8.

Identifying the time gaps is a preparatory step prior to classification. It is a necessary step because time marks are used to properly identify the breaks between the signal traces. In our dataset, the seismograms had only gaps as time marks, therefore, we only needed to identify traces as necessary objects. To receive the accurate mean measurement values, we indicated time gaps from −22 to −20, to ensure that smallergaps are included, Figure 9. After the time mark width was defined, the signals were prepared for classification. The traces were identified based on the analysis of areas of the colour contrast in a monochrome scale and recognised the position of traces. We considered the curvature of traces and measured the time gaps according to the path direction. As a result, vertical time gaps were identified for which the threshold number was set in a such way that it was enough to include the individual traces with smaller (or less distinguishable) gaps. Once we identified the time gaps, the seismograms were ready for the next steps of classification.

#### 2.5.3. Classification

The aim of the classification algorithm is to discriminate the traces from the time marks and noises, and to perform a complete object recognition, that is, all objects are to be classified and assigned to the correct category of classes. The image threshold for classification was set as 82, Figure 9, representing the minimum intensity value of a pixel to be included in classified objects. In our case, 82 was appropriate number for moderately clear images and thin lines. The image threshold minimized the number of misclassified pixels compared to other values, that is, the distributions of the grey level values in pixels is best distinguished by these values, which makes up the object of seismic lines well separated from the background of the seismogram. After we compared other values of the image thresholds, we noted that the level of 82 well separates the lines of seismic signals (as foreground pixels) from the scanned paper as background pixels. Thus, the empirical selection of this value improved the results of image processing.

The threshold parameters for seismogram classification in DigitSeis includes several values. The image threshold is set to 82, as explained above.

The time mark width indicates the distance between the tiny marks representing minutes. Since they vary in different seismograms due to the individual paper records of each seismograph, the width is detected manually on the screen. The lowest value was selected to ensure that we do not skip the possible cases. The time mark offset is an option useful for the seismograms where traces are interrupted each minute by an offset time mark. In our case, we did not have time marks; instead the minutes are indicated by gaps repeating each minute as tiny pauses in the recording. The object thickness represents a natural width of the line recorded by a drum, which is visible on the paper and detectable by a computer vision algorithm. In this case, we had a value of 25 pixels, Figure 9.

The Classification was performed using the ‘Classify Objects’ tool of DigitSeis where the objects on the image were classified into 2 categories: ‘traces’ and ‘noise’. In our case, we did not have time marks, but time gaps instead. In some cases, noise objects were misclassified into the ‘trace’ category and vice versa. Therefore, their categories were adjusted manually and corrected to the appropriate classes using the ‘Change Classification’ function. Here the traces separate the segments in a seismogram interrupted by the time gaps.

These time gaps identify minute intervals in traces which are necessary to calculate timing. The noise includes all the irrelevant objects and annotations on the seismograms, e.g., spots, handwritten texts or annotations, usually placed on the edges of seismograms. Using ‘the Classification’ tool, we converted the image from the raster TIFF format into the vector binary format and classified scanned image into a set of objects with the two main categories: (1) traces; (2) noise, see Figure 10. This dataset did not have time marks, but time gaps instead; otherwise we would have a third class of objects—‘time marks’. The classification algorithm performed the partition of the image into classes and assignment of pixels into each categories, respectively.

The algorithm of classification identified lines and traces, and discriminated them from noise and time gaps. The results of the classification include two types of objects: traces and noise, coloured by red or white, where red signifies noise (not used in the vectorisation) and white—traces (signals of the ground motion, i.e., the majority of objects on the seismogram), Figure 10. Combining separate segments of the traces was performed using the crosshairs, Figure 11b. The classification settings were adjusted using the Classify Objects function of DigitSeis, in order to make the automated processing smooth and accurate (Figure 11). In this way, the procedure of image processing generated a pixel matrix for the traces by scanning the whole image.

The analog seismograms almost always include imperfectness on the old paper, e.g., handwritten annotations, gaps in data, damaged parts of the records. Other issues arise from the technical nuances caused by the seismometer instrument and the process of signal recording. Therefore, to ensure the detailed selection of objects, a closer analysis of the selected parts of the seismogram was performed using the ’Small Region Analysis’ function, as shown in Figure 10b. As a result, traces, time gaps and noise were all classified more accurately as a prerequisite for the next step of digitising.

## 3. Results

### 3.1. Vectorisation

For vectorisation of lines, DigitSeis uses a mathematical algorithm of the ’golden-section search’ which is based on the nonlinear optimisation approach. The principle consists in finding an extremum (min/max) of a function inside a given interval [105,106]. Thus, it lessens the amplitude of the first derivative of the combined trace according to the relative vertical shift between the seismic trace and the time gaps [96]. The vectorisation was obtained using the embedded algorithm of DigitSeis with selected examples of digitised seismograms shown in Figure 12. Figure 12a shows the vectorisation for the whole seismogram, where many segments were identified correctly, while others required additional interactive manual corrections as encompassed in yellow boxes. Figure 12b visualises the enlarged fragment of the same seismogram with automatically identified traces with curvatures and time gaps.

To make the digitising workflow more effective and accurate, the images were zoomed for the enlarged analysis. The example of the processed image taken on 9 January 1954 (UCC19540109Gal_E_0812.tif) contained 45 traces with numeration from top to bottom, each horizontal line corresponds to the 30 min of measurements, Figure 12. Thus, the whole image shows seismic situation for 24 h taken during 9 January 1954 (for the case of UCC19540109Gal_E_0812.tif).

The complete processing of each seismogram required ca. 30–40 min for each image. However, the time varies individually with the most time-consuming manual tasks including the individual adjustments. These are required for each paper when dealing with noise issues, e.g., detection of noise signals and deleting them from the image. Manual corrections tools required additionally approximately, 10–15 min for each case, which also varied individually depending on the complexity of the scene. Another issue is a check of the width of time gap which was adjusted for each seismograms manually. Based on our experience, manual processing required time and the complete procedure took about 40 min for a seismogram. As a result of the semi-automated data processing, the seismograms were vectorized accurately and converted to the .mat format. To minimize this time and to improve the process, more machine learning components could be added for the next versions of the program, as a recommendation of the improvement of software. This could include, for instance, a better detection and exclusion of noise signals such as automatic recognition of the annotated handwritten notes which are often present on the old paper-based scanned seismograms.

### 3.2. Post-Processing: Correcting Traces

The majority of traces and time gaps were well identified using the DigitSeis in a semi-automatic mode, except for the selected lines that required manual corrections in an interactive regime. Some of such traces on the images were classified wrong. Figure 13 illustrates how the discriminability of trace lines against the background selected by the golden-section search is performed using the method proposed in DigitSeis.

In most of the cases, they were located in the difficult segments of the image where the program could not recognise selected traces correctly using algorithms of machine vision. These difficulties were caused by the entangled path of the lines that could not be discriminated automatically and separated from noise and resulted in obvious misclassification cases that were corrected manually. Figure 13 shows the cases where the machine could not recognise the correct path of the line. This resulted in erroneous digitising and required manual corrections of the parts of images that contain gaps, shows as yellow boxes in Figure 13.

Correcting the traces in each seismogram was performed using the ‘Correct Trace’ mode of DigitSeis. The identified traces were examined in an enlarged view (Figure 14) and the lines were then manually adjusted. The misclassified traces were corrected semi-automatically and the image was re-digitised. In such cases we identified the location and the problem of the erroneous seismic traces on an image and classifies them anew in the segments. We used the magnifying tool for enlarging a small portion of the trace in a reclassification window, Figure 14 and Figure 15. First, we identified traces that required correction, and then corrected the line accordingly. Afterwards, the updated trace information was integrated to the whole line of trace in each case. Some of such segments were manually changed in the misclassified traces and the types of the objects were corrected in these cases, e.g., annotations on the edges of paper have been assigned a category of ’noise’. Visual inspection was performed upon the re-classification.

A certain limitation is that our method using DigitSeis does not consider overlapping cases of the seismic traces with high level of signals where amplitudes of the trace lines may exceed the width of the 30-min track in a paper space, and thereby may confuse different steps of vectorisation that should be adjusted manually. For example, the cases demonstrated in Figure 14 have the two overlapping traces where the algorithm of DigitSeis could not recognise the line correctly and the line was interacted manually with vector segments corrected by hand.

Other misclassified segments are presented in complex regions of seismograms with overlapped traces where the corrections required manual adjustments. Graphical representations of the encountered problems and complicated cases are illustrated. Since correction of these cases is a time-consuming process, without taking account for overlapping neighbouring traces the processing of larger dataset becomes difficult. This may confuse the interpretation of the seismic signals and result in misinterpretation. Consequently, accurate vectorisation requires automation for more distinct recognition of the skeleton of the target trace lines. In our future work, we are going to investigate the modelling of trace lines in seismograms using algorithms of Python in order to facilitate the recognition task.

The vectorised traces were then highlighted by random colours and corresponded to the traces in the image, Figure 16. The vectorised curves of traces were optimised by the approximations with the minimum distortion of the line by spline functions. The results of the processing of the classified image Figure 17a are presented as a digitised vector image with the distinct coloured traces and minute time gaps randomly coloured on the digitised image, Figure 17b. Correcting the classification of traces was a necessary step before vectorisation.

In general, the automatic tracing of lines by DigitSeis performed well for the segments with low amplitude of seismic signals that do not cross each other. For the cases where traces overlap and interest due to the seismic-phase arrivals, large amplitudes cross the neighbouring traces. In this cases we corrected the truncated waveforms manually, as explained above. We used special tools allowing to pick up the trace, correct its vertical extent, and separate it from the neighbouring lines by manually removing the parts of the segment that were erroneously classified to the given trace. Using zooming, we isolated the segments of the seismic lines and repeated the classification process, followed by the time identification and vectorisation.

Manual correction during the vectorisation process was performed by tracing down a line in a selected segments where merged regions were separated manually and lines directions were corrected along the necessary fragments of trace. Selected polygons were assigned manually by vectorising using modality in the ’Correct Trace’ function, Figure 15.

### 3.3. Timing and Converting

After all the traces were digitised, the time was calculated using the ’Calculate Timing’ function, Figure 18. The time calculation properties were setup using a menu, where the program asks for the default time and time increment. In the demonstrated case, it is on 9 January 1954, Figure 18a. Thus, we defined the absolute time for one record as a reference point which was then highlighted in red. The basis of setting the reference time point is according to the metadata presented in each seismogram.

When originally recorded, the analog seismograms included the information regarding the date, hour and minutes of the start of seismic recording. This original information of time series data from the digital records of Uccle station was copied and used to set the reference time. For the setup of time marks, we identified the time gaps that correspond to the beginning of each minute and provided the absolute time using a special menu, Figure 18b. It included the time in HH:MM format and year-month-day period that exist in each seismogram. The time and date associated with the time mark were then entered, and the time marks were recalculated, Figure 18c.

The duration for the most of the lines was 30 min, and the seismograms were recorded by a drum of the seismometer in one day. So, we indicated the time between the gaps in seconds, and the program drew the gaps (visible yellow vertical dashed), Figure 18c and in the enlarged view, Figure 19. The time markers were visualised using the time gaps classified in a previous step and measured automatically using a threshold. Afterwards, the resulting data were exported into the MATLAB format, and saved as the .mat files. In MATLAB environment, the vector files were opened, checked and overlaid over the scanned TIFF image, respectively. The resulting files converted into MATLAB contained vector lines as traces, that is, included information received in DigitSeis. The converted file included the vectorised image with traces, its classification structure with segments and parts of the lines and time gap breaks. Therefore, a special value of the DigitSeis consists in its compatibility with MATLAB and possibility of further reuse of the processed files in other tools, such as Python. In this way, the vectorising workflow that we performed using DigitSeis is a continuous procedure can be smoothly integrated and exported to another software as a stepwise chain of the separated processes.

### 3.4. Validating theResults UsingPython

To evaluate the discriminating functionality of the seismic traces by DigitSeis algorithms over the dataset, we validated the results and performed the quality control of the digitised traces using Python [107]. The Matplotlib library [108] was applied for export and graphical visualisation of the MAT files. For tuning the parameters of the digitising process in the next series of recording, we performed the post-processing of the result using graphical visualisation and the analysis of the performed vectorisation by exporting the files in MAT format into Python, Figure 20 and Figure 21.

The length distribution of the segments imported is represented in Figure 21, the peak correspond to the typical frequent segment length $S_{typ}$ (approximately 1415 pixels in our setup for 59 s of recording). We arbitrarily keep as valid the segment inside the range $\left[0.8*S_{typ},1.1*S_{typ}\right]$. The starting position of these segments is represented as the green dots in Figure 20a. From the distribution of the short invalid segments, as shown in Figure 20a, we can identify small segments corresponding to the typical hours marks length $H_{typ}$ (around 210 pixels here). The valid hours marks are considered in the range $\left[0.9*H_{typ},1.1*H_{typ}\right]$. The respective graphs are presented for the seismogram UCC19540311Gal_E_0727_280.mat taken on 11 March 1954, Figure 21c,d.

The starting position of the hours segments is indicated by the blue dots in Figure 20. Most of the hours marks are aligned, because of the rotation speed of the main drum is two per hour. Some marks have the same length which is also presented in the border and noise segments. By looking to the colour dot distribution, it is possible to check the overall quality of the detected segments.

The misclassified pixels were induced by the distortions of paper which caused the erroneous recognition of pixels in the edge of paper (red dots). Such pixels are classified as noise in the segments of trace in a way that it erroneously depicts the edges of seismic trace, as shown in Figure 20. The average length of segments was divided automatically by DigitSeis, and then evaluated and visualised by Matplotlib, Figure 21.

Seismic traces and short time gaps were classified based on their geometrical characteristics and threshold. Therefore, the quality of scanned paper may affect the automatic recognition and interpretation. For instance, some segments were merged into one continuous line, while others were broken apart. Figure 21 illustrates the analysis of the performance of DigitSeis in vectorising the segments of a trace for the two examples (UCC19540109Gal_E_0812.mat and UCC19540311Gal_E_0727_280.mat): the average length of the identified segments of seismic lines as ’trace’ and the frequency of the segments per hour, respectively.

Seismic lines were separated into the segments, according to the traced lines (a 30-min trace on each seismograms) using algorithms of DigitSeis. The program searched for the correct path of the line and zero-lines using the defined parameters of the skeleton of line. Separating the traces and noise from the time gaps and the background on the paper was performed automatically. Therefore, screening and examining each pixel of the raster image was performed through the defined classifier thresholds which resulted in variations of selected segments of the traces.

The parameters for image classification included image threshold at 82% and the time mark width for gaps as −12 (in some cases up to −22, depending on the quality of scanned paper). The setup parameters included threshold and intervals of the time gaps corresponding to the minute marks made by the drum of the seismometer. Some gaps were identified better, while others were blurred, Figure 20. As a results, the discrimination of the traces varied based on the admissible properties using image thresholds, as assessed by Python, Figure 21.

## 4. Discussion

### 4.1. Summary

We have introduced a classification driven approach of DigitSeis for processing analog seismograms from the collection of ROB. Using the functionality of this program, we presented a workflow on dataset processing with seismograms containing time gaps. Object features, automatically detected on seismograms, were extracted from raster images using algorithms of image classification by DigitSeis. The derived vector file contained classified objects such as segments of line traces, noise and time gaps. The algorithms of DigitSeis processed well relatively large files (each scanned seismogram has a size of ca. 350–400 MB) and converted these data into .mat files.

The process of classifying and digitising paper-based seismograms is based on recognising the features of the scanned objects such as lines and handwritten marks classified as noise. The major principle of DigitSeis consists in classifying the training lines (traces of seismograms) based on the decision according to the threshold test. For instance, we set up experimentally the time gaps from −22 to −12 to indicate the small distance breaking the minute marks, which was used to identifies all those gaps throughout the image. Based on the setup parameters of threshold, the objects on seismograms were identified stepwise on the whole image.

This workflow classified the image and digitised the objects in a semi-automated way. The results were converted into the .mat format. However, certain segments of the lines required manual tracing the paths and corrections, as demonstrated previously. The most complicated cases of the overlapped traces were corrected separately using visual characteristics of these segments to discriminate the traces from noise signals. Such areas were zoomed as the selected enlarged areas on the seismogram and corrections were performed manually using a crosshair. Specifically, we identified broken segments, crossed directions of the lines and overlay in traces. using the described workflow we vectorised the seismograms from the archives of the ROB and converted them from raster into vector format.

We presented a practical application of DigitSeis software for digitising scanned historical seismograms in a semi-automatic regime. We note that while using a good digitiser is a major task of any seismological research, there are still more algorithms that need to be adjusted for a fully automated system of classification and digitising of seismograms. Processing seismograms is an essential part of geophysical research having its specifics compared to simple images: minute time marks made by the drum, seismogram timing, recognition of line directions, smooth data conversion, etc. Our goal was to present here a practical application in signal processing by DigitSeis using dataset on historical seismograms with time gaps.

### 4.2. Limitations

A limitation of our work is that we need to label all the attributes (time gaps and traces, deleting) in each seismogram, specify the noise on the background paper, check the segments for misclassification issues, correct the overlapped traces and control the identified traces manually. This is a time-consuming process and we need to acquire this for each processed seismogram. This limits the speed of data processing in case of big data issues and overall limits the scalability of our method for novel dataset. In this paper, we have proposed a DigitSeis solution to perform the vectorising task by converting scanned images in raster format into vector format. Since the output format was compatible with Python we processed data using Matplotlib library as a separate step. For a fully automated data processing and rapid building of big data corpus of digitised seismograms, we propose to use the advanced solutions of ML techniques in similar future works. As an example for this, we suggest a ML solution of combining DigitSeis with Python.

Using a fully automated methods of data-driven algorithms, for instance by Python, may improve the performance of vectorising model without human-specified manual adjustments of data processing, as was the case in our work. Therefore, one direction for our future work is to use the programming techniques to apply the algorithm of vectorization using learned attributes of seismograms (time marks/gaps, width of vector traces) for a big corpus of data. This would improve the speed and effectiveness of big data processing which is the case of large archives and libraries, such as ROB. Thereby, one can reuse the algorithm using attributes of seismograms to detect similar features on a new dataset without time-consuming work on identifying and labelling attributes. Thus, machine learning techniques of programming should be developed to adaptively determine the optimal parameters in the seismograms using attributes selected as model.

### 4.3. Future Directions

In the future work, we intend to increase our database with scanned images from other periods, due to the importance of the archived seismograms. Past seismicity of the Earth, climate modelling, exploring the atmosphere-ocean-solid earth interactions, are some of the examples of many cases and applications in geosciences where scanned historical seismograms can be used. Archives of seismological observatories containing records from the pre-satellite period are a crucial source of information for such tasks. To fully utilise the potential of historical seismograms, the abilities and advantages of modern digitising systems should be used. In our research we demonstrated the functionality of DigitSeis applied for the vectorisation of old seismograms collected from the historical dataset of ROB. Currently the archive includes data for 1954. In the next steps of our project, we will include other periods and coverage of seismograms to enlarge the database.

## 5. Conclusions

Earthquakes are one of the most hazardous geological processes which significantly affect the environment, human lives and property. The analysis of ground motions of the Earth is an essential background for prognosis of earthquakes. However, a thorough analysis of seismic data requires the use of the advanced methods for effective and robust processing and accurate interpretation. This is possible using semi-automatic approaches of signal recognition and interpretation, as we demonstrated in this paper. Analog historical seismograms are valuable source of information for retrospective modelling of the Earth’s seismicity which should be converted into a digital format, to be reused in modelling. Once digitised, a corpus of the vectorised seismograms can be effectively used for earthquake engineering, retrospective data modelling and prognosis or similar geophysical tasks. For instance, integration and comparison of old and current seismic situation enables the computational analysis of the seismicity of the Earth.

Computational geophysics benefits from the large archives of seismogram datasets which contain a valuable source of information that can be reused due to automated digitising. The subsequent generation of the digitised corpus on seismograms enables to restore seismic information contained in historical records from UCC with the aim of capturing and analysing characteristics of traces and parameters of ground motions. If scanned seismograms are digitised rapidly and accurately, they can be released in an accessible manner for further analysis. To this end, a copious amounts of historical seismic data must be processed, classified, digitised and annotated. To meet such challenges of modern seismology, the effective methods of processing analog seismograms are required. Some of the existing approaches in vectorising seismograms are presented in the proprietary software and can only be achieved through the restricted access. These methods have their advantages and shortcomings, and the use of these software is not at the level required for rapid and accurate processing of analog seismograms. Reasons for this are a lack of fully developed algorithms, finely adjusted to digitise specifically seismic data. Another reasons is that the digitising software for seismic signals processing may not be available in open access.

In response to these needs, this paper demonstrated the successful application of DigitSeis for semi-automatic vectorisation of the archived scanned seismograms. We described the workflow and pointed at the major advantages and drawbacks. The approach of DigitSeis implemented processing seismic traces and detecting time gaps based on the threshold parameters of lines. The algorithms considered specific format and characteristics of seismic data such as amplitude of traces, time gaps and 30-minute record intervals in data recording. Using algorithms of DigitSeis we converted raster scanned files from ROB archives into digital vector format for further reuse. As a central added value of this paper, we demonstrated that this state-of-the-art software is a very useful tool for digitising signal seismic traces. The algorithms of DigitSeis were used for processing a corpus of data collected from the historical records of UCC station recorded by the Galitzine seismometer in ROB, UCC, Belgium. Currently the dataset includes scanned historical images from 1954, however, the data collection with be expanded to cover other periods. The workflow of DigitSeis was used for semi-automated training of parameters of seismograms and the computer vision algorithms were applied to detect features of seismic traces. The algorithms of DigitSeis were evaluated on vectorisation task with a case of scanned monochrome seismograms of Galitzine seismometer. The signals of seismic traces were vectorised and recognised using defined thresholds and parameters of time gaps, and converted to vector format as .mat files.

Experimental results showed that DigitSeis presents a useful tool for vectorisation of analog scanned seismograms. The proposed framework is applicable to a variety of similar tasks in trace recognition for seismic datasets. For instance, the evaluated workflow for digitising seismograms can be exported to larger datasets and cover seismic archives since 1910s up to now to extend building a corpus of digitised seismograms. However, our experience also illustrated that the limits of this software which makes the fully automatic analysis impossible. For example, special cases (noises or stains, crossing lines), required a thorough manual assistance because the functionality and operations in vectorising were limited and required interactive corrections of the recognised traces. Therefore, the tools of DigitSeis still require improvements for the processing of big data as streams. Nevertheless, the Integration of workflow with ML is possible if the generated file are converted into the .mat format for post-processing in MATLAB or Python. Data integration enables overlay of vectorised traces over the scanned images for data control. The advantages of DigitSeis in digitising seismograms enable recurrent and consistent information capture on seismic signal levels, repeatability of the recorded peaks, time gaps, period of data capture, intensity of signals and strength of Earth’s ground motion. All this information can be used for seismicity analysis and relevant geophysical studies.

## Figures and Tables

**Figure 1 sensors-23-00056-f001:**
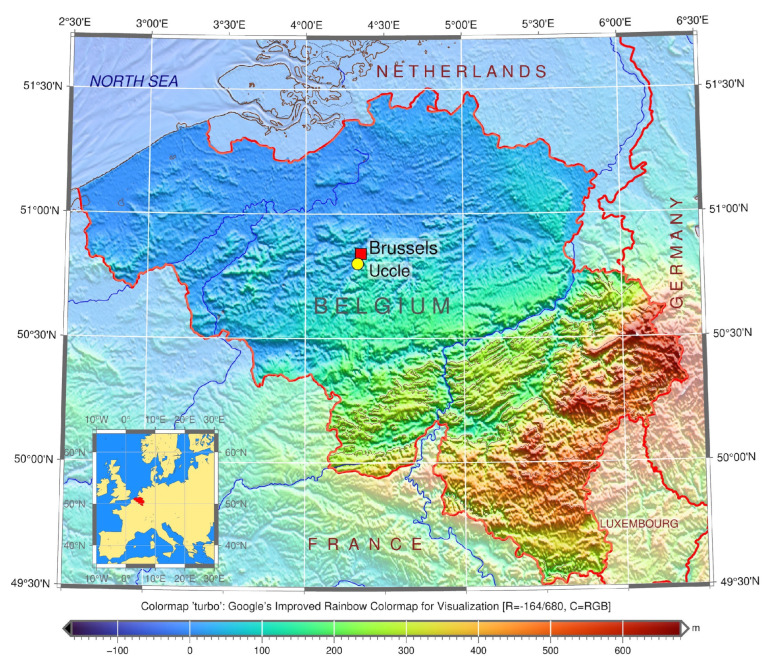
Study area: Location of Uccle seismic station, ROB on the topographic map of Belgium. Software: GMT version 6.1.1, creator: P. Wessel et al. [97], University of Hawai’i at Mānoa, Honolulu, HI, USA. Cartography source: authors.

**Figure 2 sensors-23-00056-f002:**
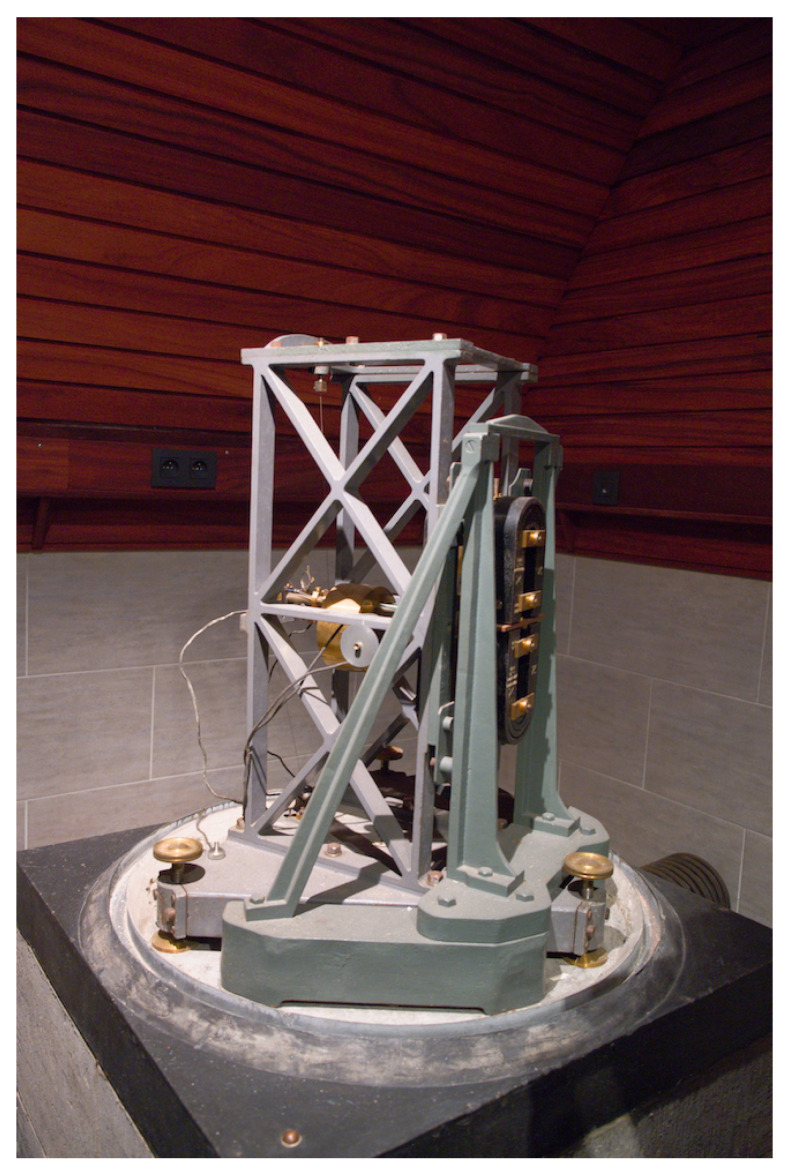
Instrument used for data capture in 1954: Horizontal Galitzine seismometer located in UCC. Image source: courtesy of ROB. Photo source: R. S. M. De Plaen.

**Figure 3 sensors-23-00056-f003:**
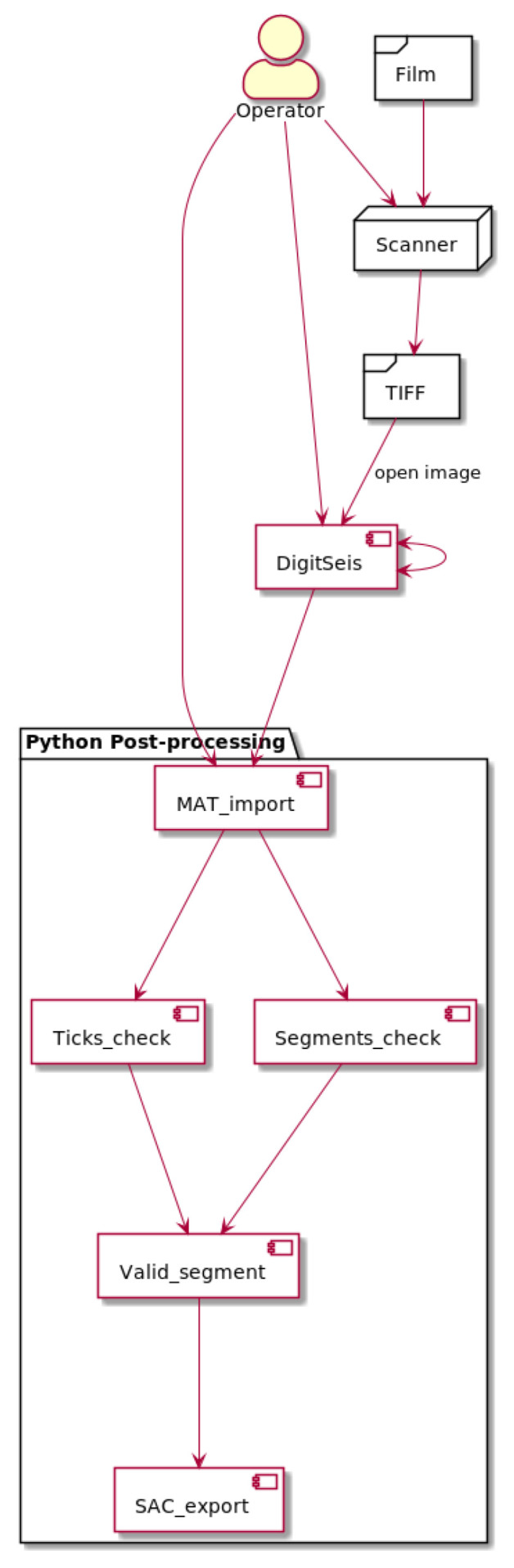
Conceptual flowchart for the technical process. Plotting: PlantUML.

**Figure 4 sensors-23-00056-f004:**
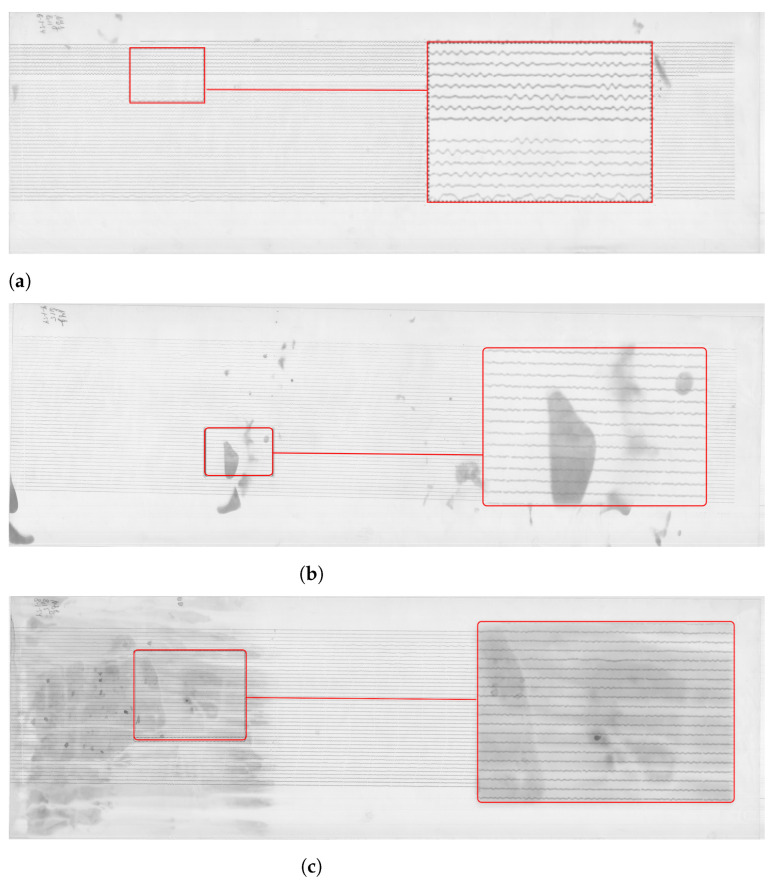
Examples of the raw data: old analog seismograms recorded by Uccle seismic station, Belgium, 1954, scanned and saved as TIFF files. Some images have visible distortions and defects. (**a**) Empty records between the lines of seismic traces with enlarged fragment of seismogram. Here: UCC19540106Gal_N_0811.TIFF. (**b**) Partially spotted image caused by storage with enlarged fragment of seismogram. Here: UCC19540107Gal_N_0815.TIFF. (**c**) Continuous noise dark background on the image with blurred traces which causes lack of contrast for the automated image recognition. Here: UCC19540108Gal_N_0815.TIFF. (**d**) Overlapped traces of a very large event which cause a problem during vectorising with selected enlarged fragment of seismogram. Here: UCC19540112Gal_E_0750.TIFF.

**Figure 5 sensors-23-00056-f005:**
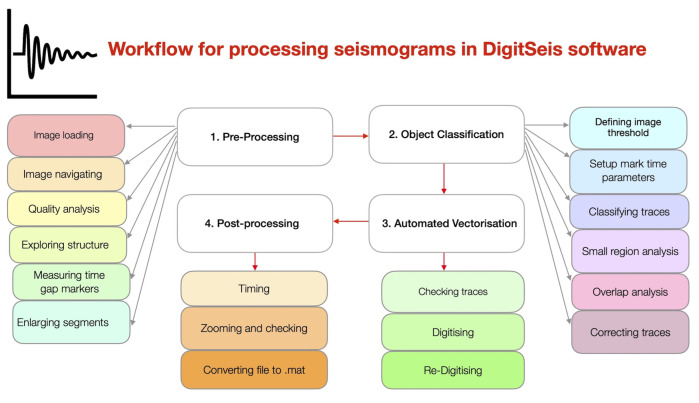
Pipeline schematic diagram for workflow process for vectorising seismograms.

**Figure 6 sensors-23-00056-f006:**
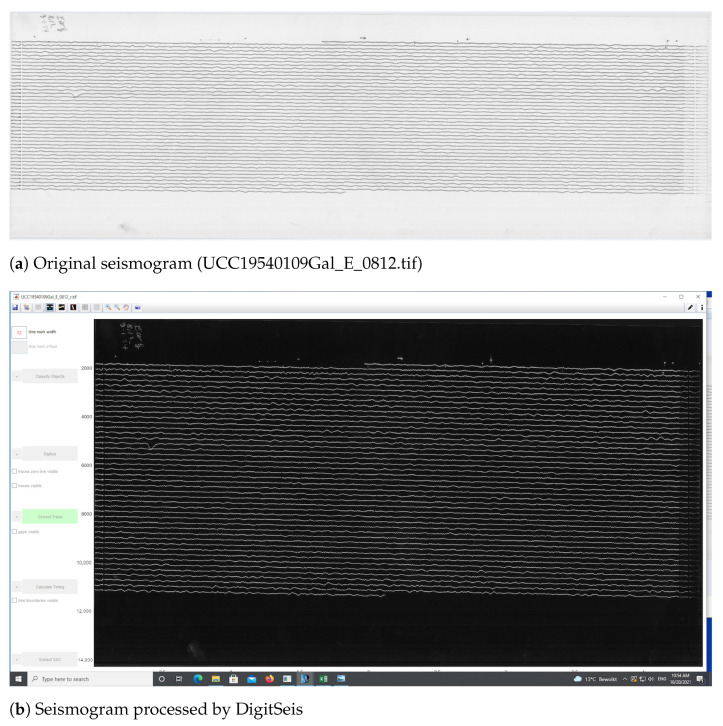
Seismogram recorded from Uccle station on 9 January 1954.

**Figure 7 sensors-23-00056-f007:**
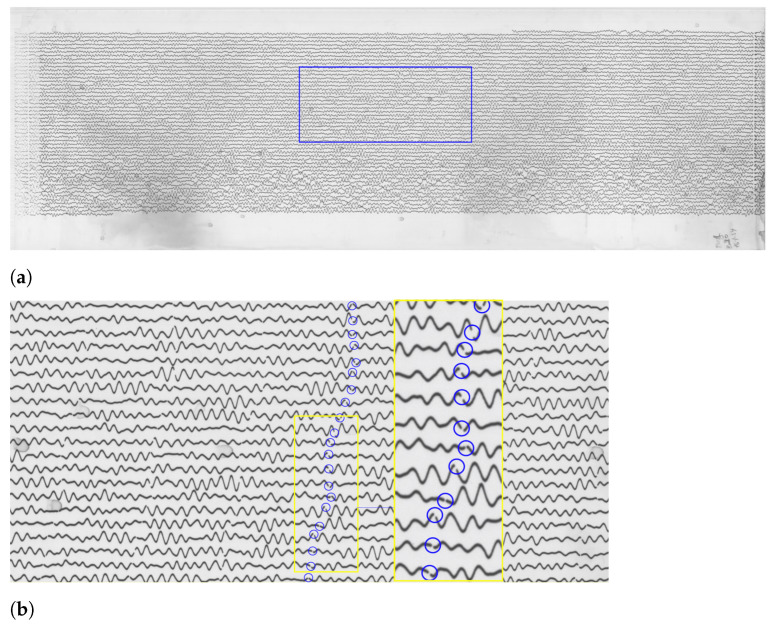
Analysing seismogram in paper-based format: traces and time gaps. (**a**) Original scanned seismogram (UCC19540116Gal_E_0820.tif). (**b**) Enlarged fragment of image showing time gaps which indicate minute marks and a zoomed segment separating the trace line between each other with tiny white gaps breaking the traces.

**Figure 8 sensors-23-00056-f008:**
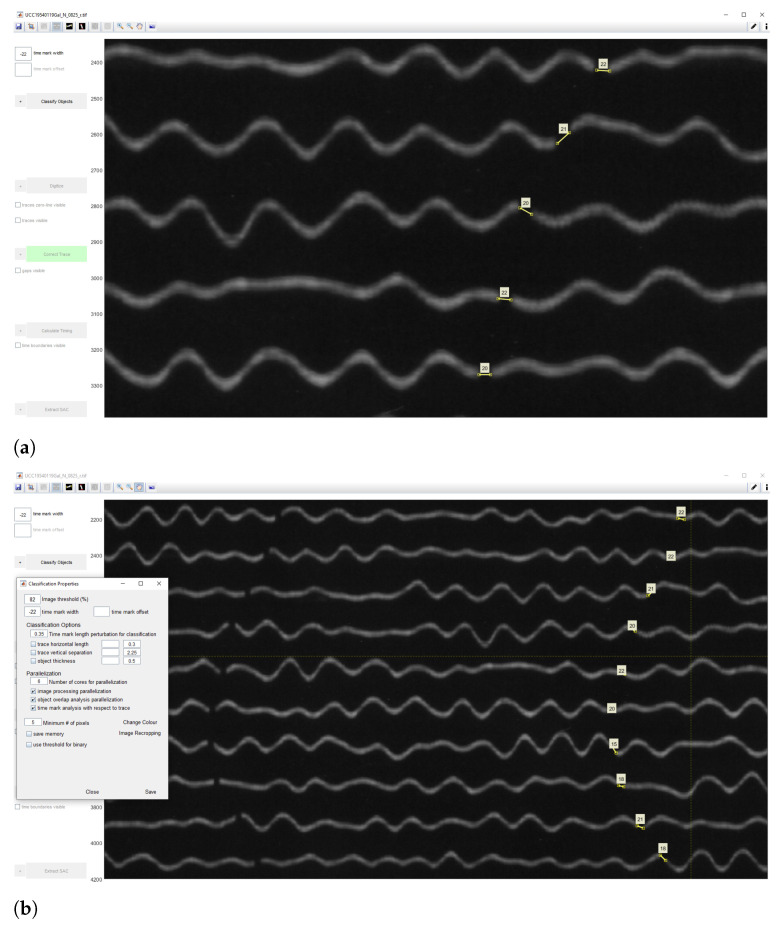
Classification of seismogram processed by DigitSeis. (**a**) Identifying the time marks on seismograms by measuring time gap between the records. Here: example on file UCC19540119Gal_N_0825.tif. (**b**) Indicating the time marks on seismograms as −22 and preparing image for the classification.

**Figure 9 sensors-23-00056-f009:**
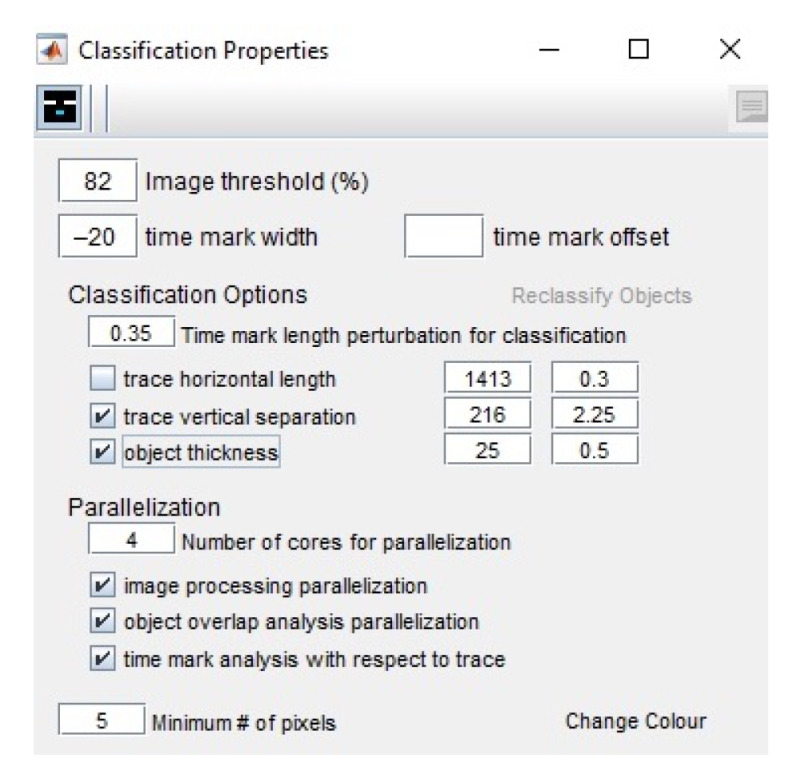
Threshold parameters for seismogram classification in DigitSeis.

**Figure 10 sensors-23-00056-f010:**
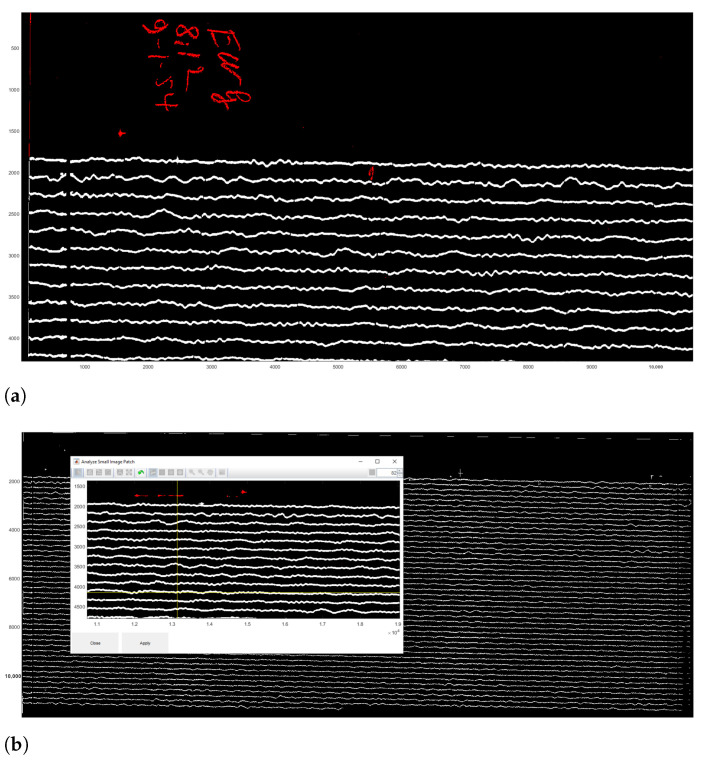
Results of the paper seismogram classification processed by DigitSeis. (**a**) Results of the classified seismogram with shown identified object categories (fragment): traces (white) and noise (red, here: handwritten annotations). (**b**) Small region analysis used for defining a smaller area of interest for closer examination of a border region of the seismogram.

**Figure 11 sensors-23-00056-f011:**
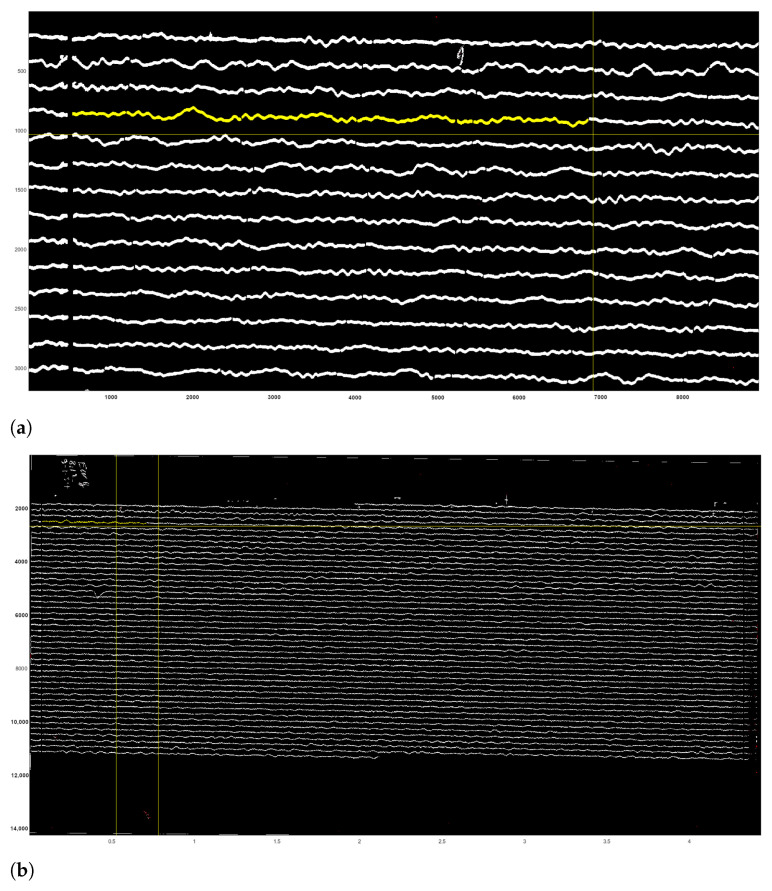
Results of the scanned seismogram classification processed by DigitSeis. (**a**) Results of the classified image with shown yellow segments of the identified trace (enlarged fragment). Here: example of file UCC19540109Gal_E_0812.tif. Yellow color signifies a selected segment on the trace line.(**b**) Classified seismogram with traces saved in binary format 0–1. Here: example of file UCC19540109Gal_E_0812.tif (seismogram recorded on 9 January 1954).

**Figure 12 sensors-23-00056-f012:**
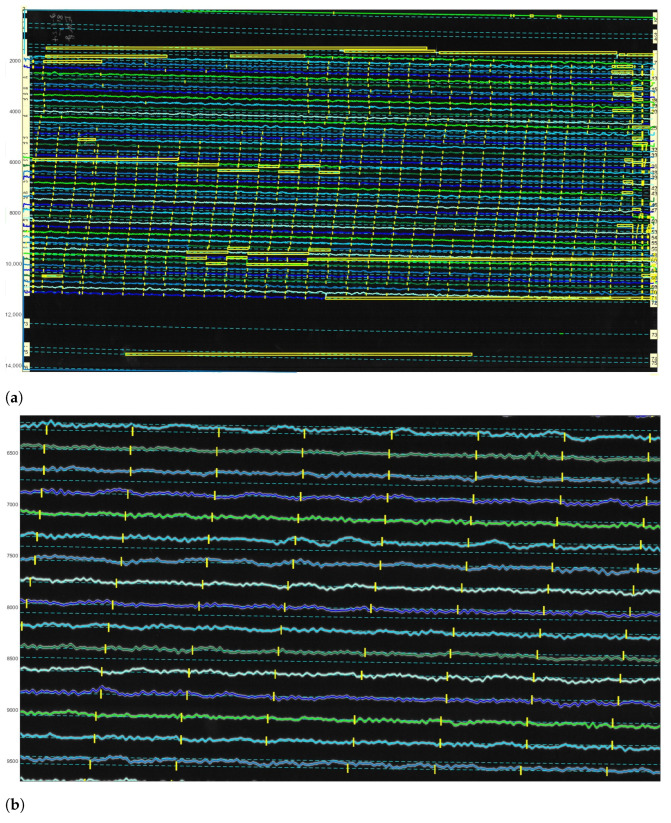
Results of the paper-based seismogram vectorisation processed by DigitSeis. (**a**) Digitised traces upon completion of the automated vectorisation. Some time gaps in the upper left part of the image were too small and not clearly visible for the automatic discrimination between the trace and the dark background. In these cases, gaps required manual correction to identify time intervals. (**b**) The enlarged view of the automatically recognised digitised traces displayed by the lines of various colours, zero-lines for each trace (cyan, dashed lines, numbered from top to bottom) and time gaps (vertical yellow dashes).

**Figure 13 sensors-23-00056-f013:**
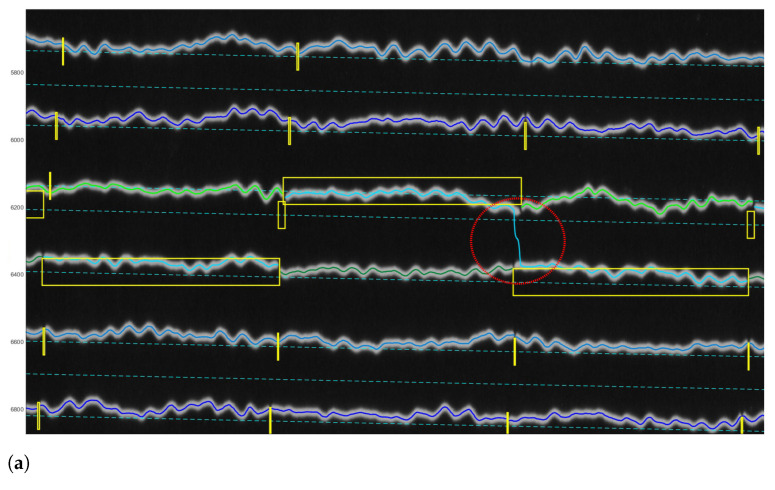
Identified traces for selective correction and re-digitising using Correct Trace mode. (**a**) Identified wrong vector direction of line crossing individual traces. (**b**) Detected misclassifications caused erroneous digitising. The gaps on the zero-lines (small yellow boxes) show the gaps that existed in the old paper in the original image itself. Differently colored lines are used to distinguish the segments of the traces.

**Figure 14 sensors-23-00056-f014:**
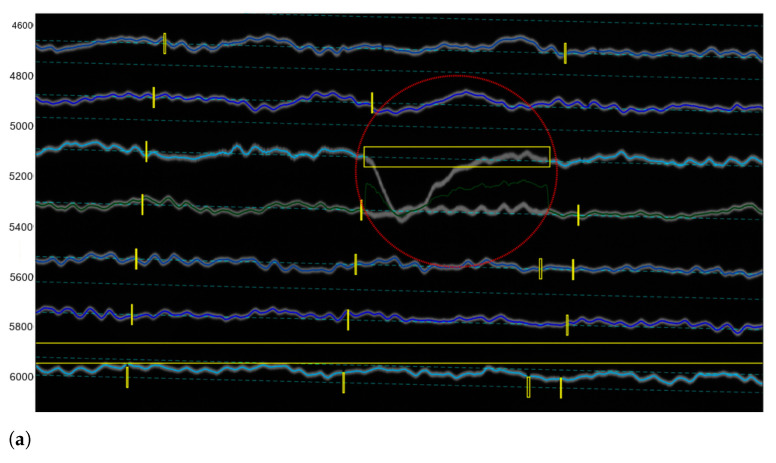
Correcting misclassified traces with wrong direction based on colour and geometric pixel’s characteristics. The pixels on the monochrome image were assigning to categories using threshold values to 1 (white) and all other pixels to 0 (black) for automated detection of lines. The differently colored lines are used to distinguish the segment of trace lines. (**a**) Overlap of line traces unrecognised during digitising: one segment of trace went steeply downwards and merged with another trace. (**b**) Enlarged view of the manually corrected entangled traces.

**Figure 15 sensors-23-00056-f015:**
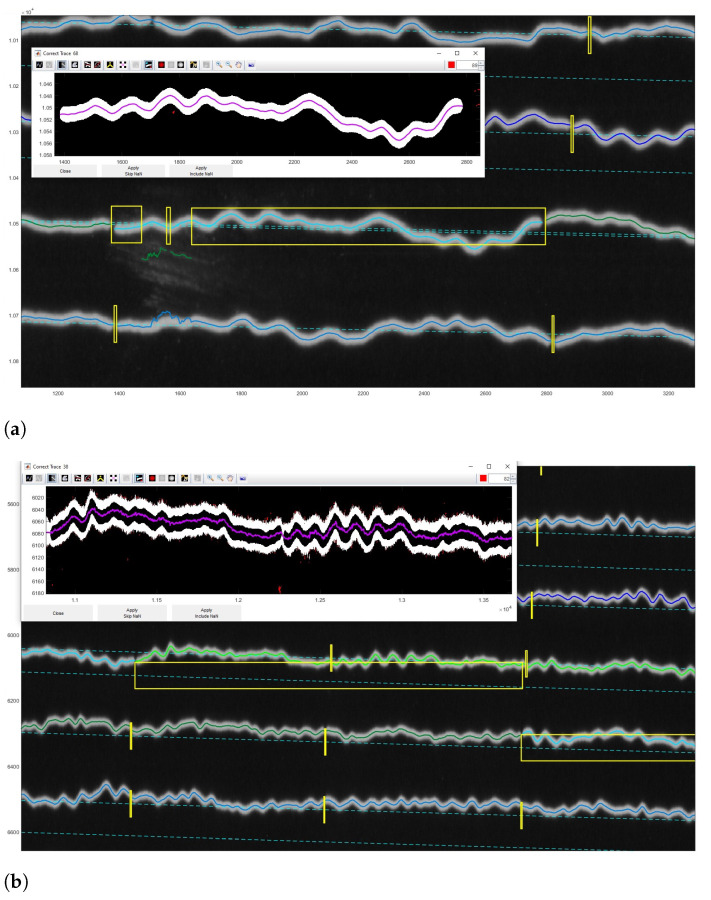
Correcting trace for the selected segments. (**a**) Merging the trace initially broken into the three separate parts (three small yellow boxes). (**b**) Reclassification of the selected segment and digitising the centroid of the trace line.

**Figure 16 sensors-23-00056-f016:**
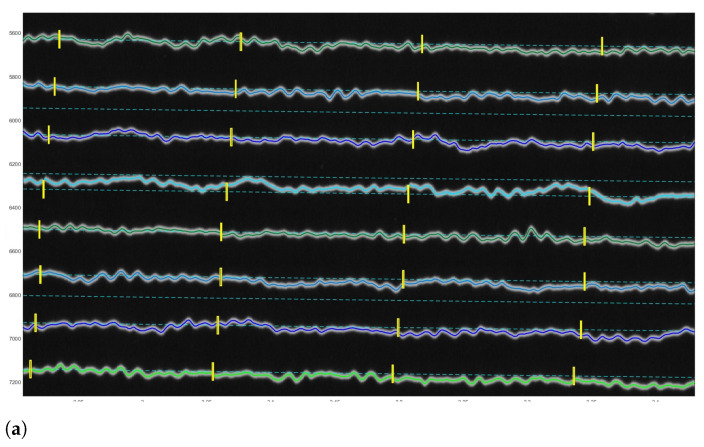
Corrected digitised traces upon completion of the semi-automated vectorisation. (**a**) Corrected digitised traces upon the completion of the semi-automated vectorisation process. (**b**) Visible SDT lines (bold) after the semi-automated vectorisation.

**Figure 17 sensors-23-00056-f017:**
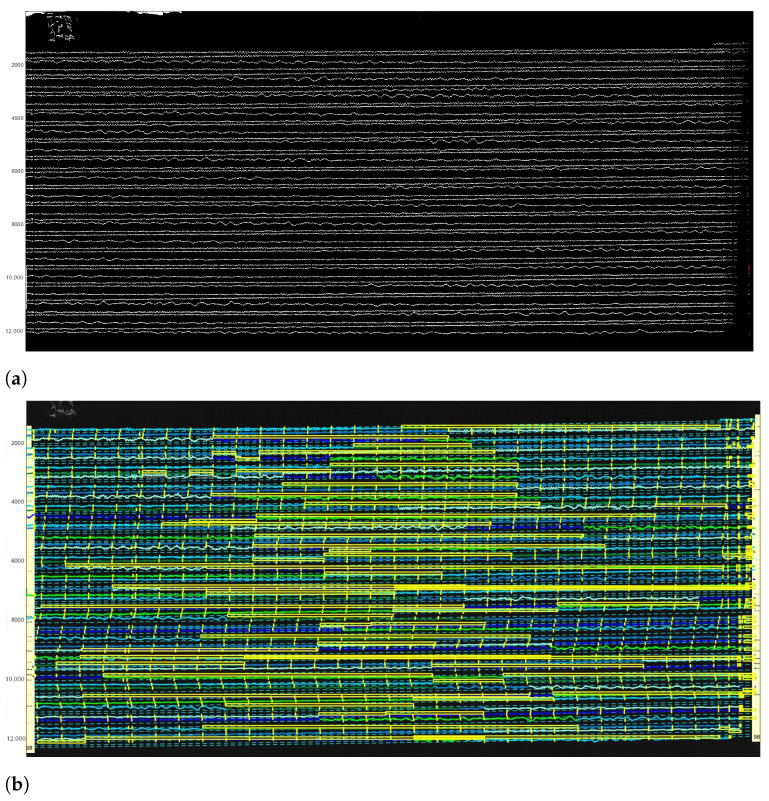
Classified image (**a**) and digitised results (**b**). Here: UCC19540311Gal_E_0727 (11 March 1954). (**a**) Classified image recognised by DigitSeis into ‘traces’ and ‘noise’ (Here: the image UCC19540311Gal_E_0727.TIFF). (**b**) The distinct coloured traces and minute time gaps are visible on a digitised image (Here: the resulted file in a MATLAB format, UCC19540311Gal_E_0727.mat).

**Figure 18 sensors-23-00056-f018:**
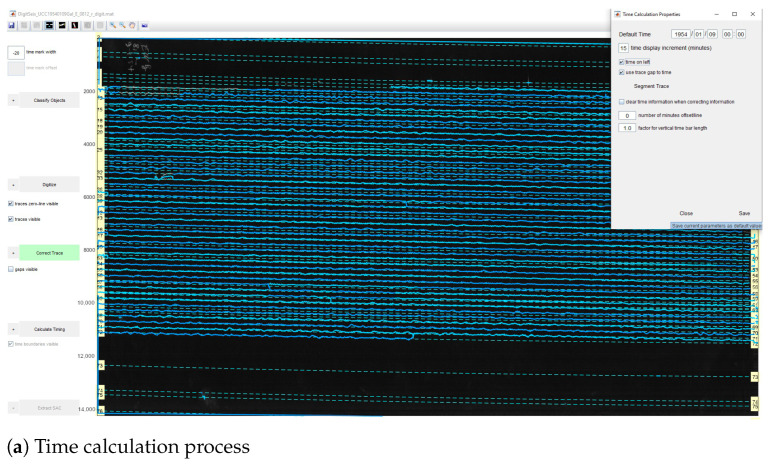
Seismogram image with adjusted timing. Here: UCC19540311Gal_E_0727.mat.

**Figure 19 sensors-23-00056-f019:**
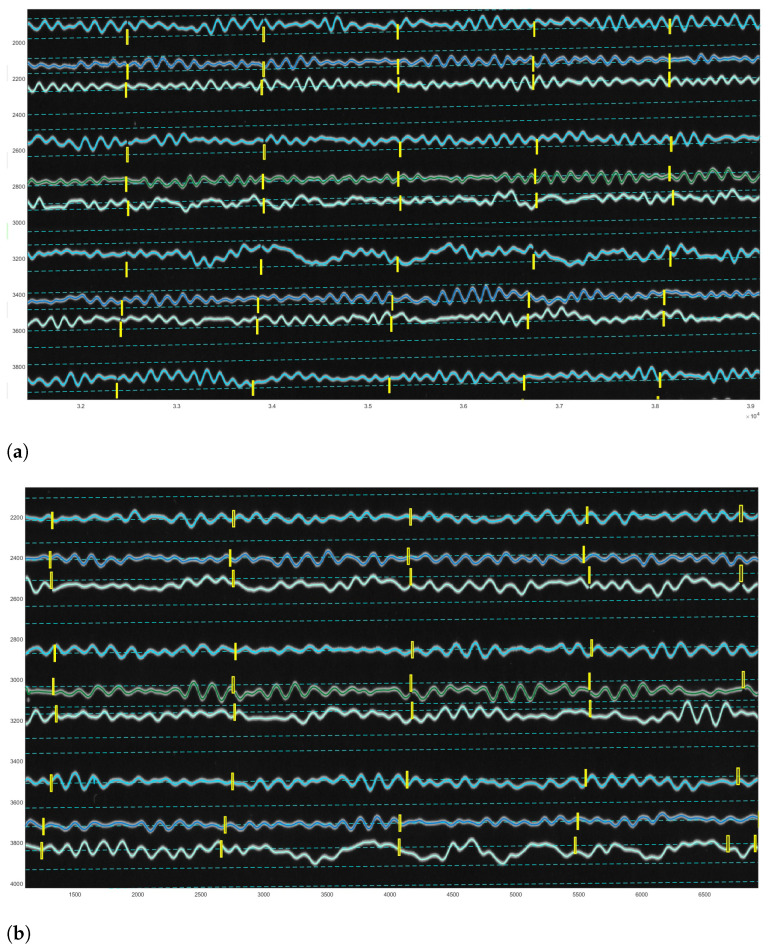
Fragments of the digitised image (Here: UCC19540311Gal_E_0727.mat) (11 March 1954). (**a**) Scanning of trace lines on the digitised image (Here: UCC19540311Gal_E_0727.mat). (**b**) Separation of traces and time gaps by filtering (time gap setup as ‘−12’ in this case).

**Figure 20 sensors-23-00056-f020:**
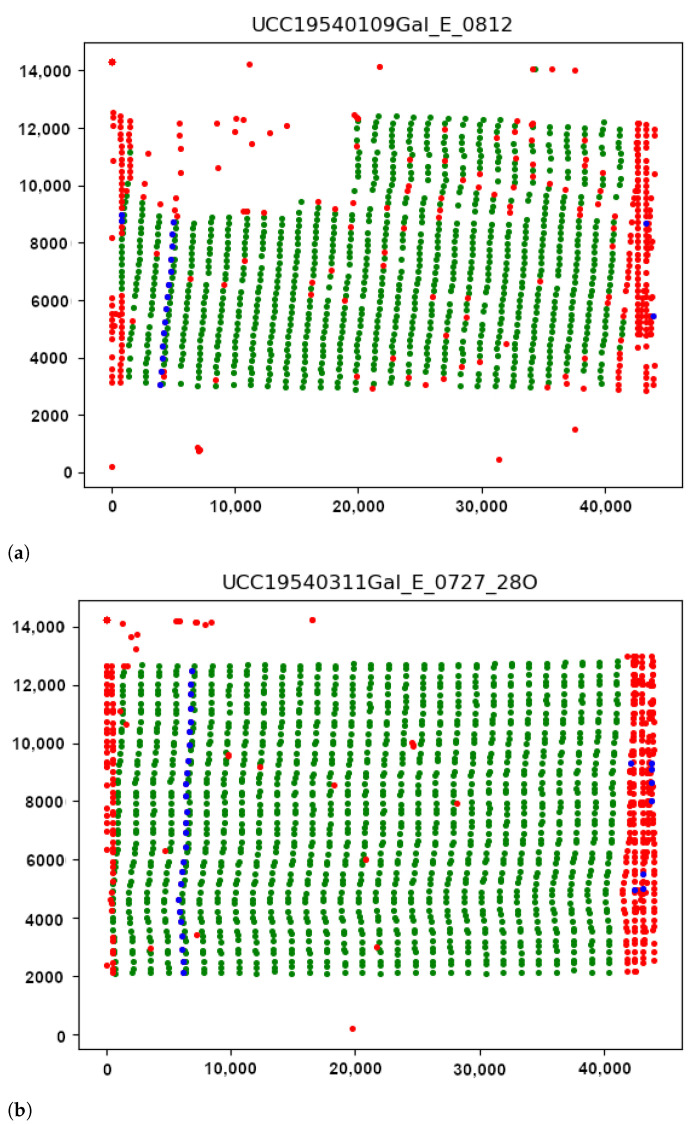
Controlling digitising results using Python (Matplotlib library). (**a**) Quality control for time gaps: missed marks in unrecognised segments. (**b**) Correctly identified time gaps controlled by Python’s Matplotlib. Different color dots are used to separate minute marks (green), hour marks (blue) and margin values and outliers (red).

**Figure 21 sensors-23-00056-f021:**
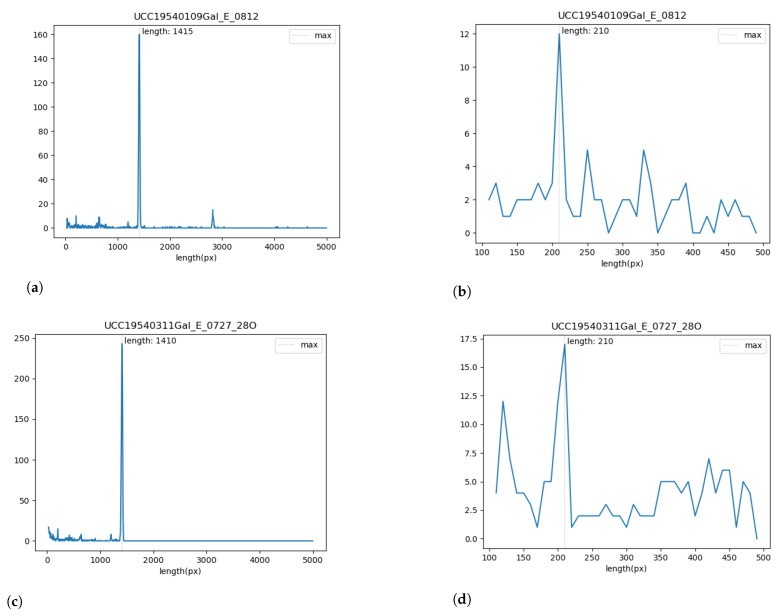
Validating the results of segments tracing in Python, Matplotlib for selected seismogram. (**a**,**c**): Frequency of segment length: size of various segments in a trace. (**b**,**d**): Analysis of statistics on segment length per hours of recording. (**a**,**b**): UCC19540109Gal_E_0812.mat (**c**,**d**): UCC19540311Gal_E_0727_280.mat.

## Data Availability

Data supporting reported results are the courtesy of Royal Observatory of Belgium, Department of Seismology and Gravimetry (OD2), used in this study as available archived datasets of seismograms. Data are available in the Cytomine workspace.

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
