# Peer review of "Computer Vision Algorithms of DigitSeis for Building a Vectorised Dataset of Historical Seismograms from the Archive of Royal Observatory of Belgium"

_sensors, 2022, doi:10.3390/s23010056_

Round 1

Reviewer 1 Report

Presented work developed during SeismoStorm project can be an useful guide for future works and projects that can use DigitSeis software and references given in your paper.

Author Response

Dear Editors of Sensors,

Please find attached the revised version of the paper. We have carefully followed the comments and suggestions of the 4 reviewers and corrected the manuscript accordingly.

All the corrections in the text are marked up yellow for Track Changes.

The replies to the comments of the reviewers are listed below.

Using the opportunity, we thank the reviewers for careful reading of the paper which improved the initial version of the manuscript.

With kind regards, - Authors.

04.12.2022.

Reviewer 1

No

Reviewer’s Comments

Authors’ actions

1

Presented work developed during SeismoStorm project can be a useful guide for future works and projects that can use DigitSeis software and references given in your paper.

Many thanks for the endorsement of our work. Yes, DigitSeis can be used for vectorization of seismic data in other projects and future similar works. We also plan to continue seismic data processing using various approaches. In our future works, we plan to compare this work on DigitSeis software with other methods in data vectorization, e.g., Python-based machine learning approaches. Thank you once more again for the review and comments on our manuscript.

Original Report Reviewer 1

Open Review

English language and style

( ) English very difficult to understand/incomprehensible
( ) Extensive editing of English language and style required
( ) Moderate English changes required
(x) English language and style are fine/minor spell check required
( ) I don't feel qualified to judge about the English language and style

Yes

Can be improved

Must be improved

Not applicable

Does the introduction provide sufficient background and include all relevant references?

(x)

( )

( )

( )

Are all the cited references relevant to the research?

(x)

( )

( )

( )

Is the research design appropriate?

(x)

( )

( )

( )

Are the methods adequately described?

(x)

( )

( )

( )

Are the results clearly presented?

(x)

( )

( )

( )

Are the conclusions supported by the results?

(x)

( )

( )

( )

Comments and Suggestions for Authors

Presented work developed during SeismoStorm project can be an useful guide for future works and projects that can use DigitSeis software and references given in your paper.

Submission Date

17 October 2022

Date of this review

10 Nov 2022 12:04:46

Reviewer 2 Report

The paper is best described as an empirical report of using the DigitSeis software to perform a digitization of historic seismograms from pre-standardization era. A great introduction is given to the overall problems related to making the historic seismology data available for analysis. I would like to point out that the authors did an amazing job of making the paper understandable for researchers in different fields, as well as describing the subject matter in such a way for readers without seismology background to gauge the problemset and the technical challenges associated with working with historical data in this domain.

The main issue I can see with this paper is a vague formulation of the results, in terms of the list of issues and problems the solution for which could advance the methodology and give information for the researchers in this field, as well as the developers of DigitSeis or the ones working on similar software, to improve the design and the algorihtms involved.

1. It is clear that the usage of the software requires manual intervention (by design of the methodology, as well as stemming from the actual experience of the authors), however it is not clear at all how much time was spent on manual corrections, what were the most time-consuming manual tasks, and what steps could theoretically be taken to minimize this time and to improve the process.

2. The analysis of specific issues with computer vision algorithms which generated problems with pre-processing, digitization, classification, vectorization, etc. - was desribed in the text, however the paper, in my opinion, lacks a clear summary of the specific issues in order of their relevance and severity, which could give an idea to computer vision specialists of what are the main issues with the process and how it could be improved. This is especially clear when the authors describe the Figure 21 - it is not clear at all what the readers should take out from this figure and from its discussion - whether the obtained results are good, bad, acceptable for further analysis, etc.

3. In several parts of the paper authors refer to machine learning tools (e.g. lines 107, 275, 556-558). In the introduction (line 107) the authors state that they are going to identify how ML tools could be used to improve the process (which, in my opinion, the authors do not do), in section 2.5 the authors claim that some specific tasks could only be solved by using ML tools without references or any kind of evidence and corroboration, and, finally, in the conclusion the authors state that the ML tools could be useful for improving the process and produce a vague suggestion that "ML solution of combining DigitSeis with Python" would help somehow. All this is very vague and does not give any insight of what is expected from any kind of ML technologies for improving the process. In order to add this to discussion the authors should clearly state the specific problems which in their opinion could be solved by ML tools, otherwise all these are simply empty statements.

A list of small typos and issues found:

  - DigitSeis software is first mentioned in section 1.4 (line 97), please add the reference citation there, as it is only cited later.

  - line 463 - "revolutions per hours" - probably meant "per hour".

  - line 540 - "very used tool" - probably meant "very useful tool".

  - lines 549-550 - sentence ".. this software limits making the fully automatic analysis impossible" - consider rephrasing, probably meant "... the limits of this software make the fully automatic analysis impossible".

  - line 590 - "available to through open access" - probably meant "available through open access".

Overall, I think that the paper would be interesting to the community, but it would greatly benefit if a clear summary of all found issues in the process were given, with suggestions of what improvements could be made in the software (from the point of the user and from the point of the properties of analyzed data), as well as more clear presentation of the results.

Author Response

Dear Editors of Sensors,

Please find attached the revised version of the paper. We have carefully followed the comments and suggestions of the reviewers and corrected the manuscript accordingly.

All the corrections in the text are marked up yellow for Track Changes.

The replies to the comments of the reviewers are listed below.

Using the opportunity, we thank the reviewers for careful reading of the paper which improved the initial version of the manuscript.

With kind regards, - Authors.

04.12.2022.

Reviewer 2

No

Reviewer’s Comments

Authors’ actions

1

Are the methods adequately described? – Can be improved.

The Methodology section is updated with more comments and explanations included in the text. The workflow is described in a more more structure way with pointed major steps. The process of digitizing is explained more detailed. Some corrections are made in the subsections 2.4. Data; 2.5. Software and Workflow, 2.5.1. Image preprocessing, 2.5.2. Identifying the time gaps. Added several paragraphs regarding the parameters of image thresholding in seismograms in section 2.5.3. Classification. The algorithm of classification which identified lines and traces is described in a more detailed and stepwise way.

2

Are the results clearly presented? – Can be improved.

The results are formulated more structurally and clearly. Some phrases are included or modified, others expanded with added more explanations (all changes are marked up in the text.)

3

Are the conclusions supported by the results? – Can be improved.

We updated the Discussion and Conclusion sections. The Discussion section is now divided into 3 subsections: 4.1. Summary; 4.2. Limitations; 4.3. Future directions. It includes a more structured summary of the presented work and highlighted the importance of of the advanced methods for seismogram vectorisation. Other two subsections discuss current limitations and the main points of future directions which will include machine learning and automation of vectorisation.

4

The paper is best described as an empirical report of using the DigitSeis software to perform a digitization of historic seismograms from pre-standardization era. A great introduction is given to the overall problems related to making the historic seismology data available for analysis. I would like to point out that the authors did an amazing job of making the paper understandable for researchers in different fields, as well as describing the subject matter in such a way for readers without seismology background to gauge the problem set and the technical challenges associated with working with historical data in this domain.

Many thanks for the support of our research. Yes, we tried to explain general problems related to the geophysical data recording in Introduction section with provided information on seismology and the nature of seismic waves, methods of their recoding and the development of the tools since 19th century up to now, and what are the current problems of seismograms data processing (automation for big data analysis and increase of accuracy which needs programming tools).

Besides, we have slightly restructured the Introduction section to make it more concise. 1.1. Background (the description of the seismicity of the Earth, associated geophysical processes, mention the development and progress of the seismogram recordings); subsection 1.2. Motivation (importance of seismograms, applications, importance of methods of accurate processing of seismogram for data interpretation); 1.3. Related Work (existing methods of seismogram processing and present many examples of data vectorization, key issues of digitizing in terms of algorithms and data processing); subsection 1.4. Contribution (our contribution to the seismogram processing, goals and objectives of this work, methods and scope of this research, and large volume of data that should be digitized rapidly using effective and robust methods).

5

The main issue I can see with this paper is a vague formulation of the results, in terms of the list of issues and problems the solution for which could advance the methodology and give information for the researchers in this field, as well as the developers of DigitSeis or the ones working on similar software, to improve the design and the algorithms involved.

The results are updated and include more paragraphs with comments on the vectorisation process, the time for processing the files and post-processing issues. The results of the vectorized lines are presented in Figures 16, 19 with added comments. The results illustrates the processed files from the large archive of the seismograms containing data recorded in 1954. Since the archive includes large amount of data, the automated processing is essential, which we used by the DigitSeis functionality, as reported in this study. Moreover, we identified some limitations and problems encountered when processing data in DigitSets, and listed them in the Discussion and subsection 3.2. Post-Processing: Correcting Traces (lines 451-457 and 458-483) where we are talking about the cases of the misclassified segments and their correction, overlapped segments illustrated in Figure 14 and time required for adjustment of the line segments.

6

1. It is clear that the usage of the software requires manual intervention (by design of the methodology, as well as stemming from the actual experience of the authors), however it is not clear at all how much time was spent on manual corrections, what were the most time-consuming manual tasks, and what steps could theoretically be taken to minimize this time and to improve the process.

We have added a detailed comment regarding time processing in section 3. Results, 3.1. Vectorisation subsection: “The complete processing of each seismogram required ca. 30-40 minutes for each image. However, the time varies individually with the most time-consuming manual tasks including the individual adjustments. These are required for each paper when dealing with noise issues, e.g., detection of noise signals and deleting them from the image. Manual corrections tools required additionally approximately, 10-15 minutes for each case, which also varied individually depending on the complexity of the scene. Another issue is a check of the width of time gap which was adjusted for each seismograms manually. Based on our experience, the manual processing required time and the complete procedure took about 40 minutes for a seismogram. As a result of the semi-automated data processing, the seismograms were vectorized accurately and converted to the .mat format. To minimize this time and to improve the process, more machine learning components could be added for the next versions of the program, as a recommendation of the improvement of software. This could include, for instance, a better detection and exclusion of noise signals such as automatic recognition of the annotated handwritten notes which are often present on the old paper-based scanned seismograms.’

7

2. The analysis of specific issues with computer vision algorithms which generated problems with pre-processing, digitization, classification, vectorization, etc. - was described in the text, however the paper, in my opinion, lacks a clear summary of the specific issues in order of their relevance and severity, which could give an idea to computer vision specialists of what are the main issues with the process and how it could be improved. This is especially clear when the authors describe the Figure 21 - it is not clear at all what the readers should take out from this figure and from its discussion - whether the obtained results are good, bad, acceptable for further analysis, etc.

The summary of the specific issues is updated and added in the Discussion and in various parts of the Results and Methodology. These includes the subsections 3.1. Vectorisation (included information about the complete processing workflow of each seismogram including time of manual processing); subsection 3.2. Post-Processing: Correcting Traces (updated comments on the discriminability of trace lines against the background and misclassification cases (pp. 15-16). Added information about the limitations in cases of high amplitudes of the neighbouring trace signals causing overlapped lines and manual adjustment of the segments – specifically, we commented that for the big data processing which is the case of archives of historical seismograms, such issues require more automation in data processing (pp. 17-18). Added discussion and some comments about the performance of DigitSeis in automated recognition of the skeleton of the target trace lines. Updated the Discussion section (pp. 25–27) which is now divided into the 3 subsections with structured information – 4.1. Summary; 4.2. Limitations and 4.3. Future directions, and Conclusion section (pp. 26-27).

8

3. In several parts of the paper authors refer to machine learning tools (e.g. lines 107, 275, 556-558). In the introduction (line 107) the authors state that they are going to identify how ML tools could be used to improve the process (which, in my opinion, the authors do not do), in section 2.5 the authors claim that some specific tasks could only be solved by using ML tools without references or any kind of evidence and corroboration, and, finally, in the conclusion the authors state that the ML tools could be useful for improving the process and produce a vague suggestion that "ML solution of combining DigitSeis with Python" would help somehow. All this is very vague and does not give any insight of what is expected from any kind of ML technologies for improving the process. In order to add this to discussion the authors should clearly state the specific problems which in their opinion could be solved by ML tools, otherwise all these are simply empty statements.

The mentions of the ML are corrected. The DigitSeis operates by a semi-automated approach which required certain corrections and human-based control during the vectorisation process. So, some phrases and sentences are modified to the ‘machine-based’ approach and algorithms of semi-automated vectorisation. We used Python in the final step of data processing for check and quality control where the vectorized file were imported externally. The next directions of our research will include machine learning components where we test the vectorisation of seismograms using Python. In the scope of this research we used the DigitSeis with a post-processing of data in Python as a separate step. So, these phrases are corrected and modified in the text (always marked up for track change control).

9

A list of small typos and issues found:

  • DigitSeis software is first mentioned in section 1.4 (line 97), please add the reference citation there, as it is only cited later.

  • line 463 - "revolutions per hours" - probably meant "per hour".

  • line 540 - "very used tool" - probably meant "very useful tool".

  • lines 549-550 - sentence ".. this software limits making the fully automatic analysis impossible" - consider rephrasing, probably meant "... the limits of this software make the fully automatic analysis impossible".

  • line 590 - "available to through open access" - probably meant "available through open access".

All these types are corrected.

  • DigitSeis is cited when first mentioned in the subsection “Contribution” of Introduction (phrase “To this end, we applied the DigitSeis software [105] for semi-automated vectorising of seismograms.”, lines 137-138).

  • Corrected.

  • Corrected in Conclusions (Phrase “As a central added value of this paper, we demonstrated that DigitSeis software is a very useful tool for digitising<...>”)

  • Corrected as suggested to "the limits of this software make the fully automatic analysis impossible", 2nd paragraph in Conclusions.

  • Corrected to “available through open access” (in Conclusions).

Besides, we have proofread the manuscript throughout: corrected all occasional typesetting misprints and minor grammar mistakes (spelling, punctuation) where necessary. Grammar errors are corrected and misprints are checked everywhere in the text.

10

Overall, I think that the paper would be interesting to the community, but it would greatly benefit if a clear summary of all found issues in the process were given, with suggestions of what improvements could be made in the software (from the point of the user and from the point of the properties of analyzed data), as well as more clear presentation of the results.

Many thanks for the endorsement and support of our paper. Yes, we tried to present it readable to the community and as clear as possible for colleagues outside the geophysical and seismological domains. The vectorisation of scanned papers is a general purpose problem, which advanced since 1980s when various materials were digitised and converted to vector format. Specifically for seismology, the vectorisation of seismograms has its specifics. This includes time marks and gaps, patterns seismic traces and the need to exclude the noise which often present in the background.

The next step includes the interpretation of the intensity of signals.

We have improved the structure of the paper: reorganized the Introduction, added many additional paragraphs and some phrases to better explain the workflow.

The comments are added in the Discussion section regarding the workflow. Here we included comments on the improvements which could be made in the software from the point of user. We commented on the need of big data, i.e., to process the data in large volumes, which are a common in archives of seismological observatories.

The Conclusion is updated with a more clear summary of the work.

We included the subsection 4.2. Limitations in the Discussion section. Here we discussed some imperfect issues of the DigitSeis, such as overlapping of neighboring lines, lack of automation component and many manual work required for adjustment of the vectorisation process. More reflections is added about possible improvement. We also added the subsection 4.3. Future directions where we point at our next steps to use the algorithms of Python for automatic vectorisation of lines.

The Results section is updated with included more comments on the obtained results.

All the new insertions are marked up yellow.

Open Review

Original Review Report

English language and style

( ) English very difficult to understand/incomprehensible
( ) Extensive editing of English language and style required
( ) Moderate English changes required
(x) English language and style are fine/minor spell check required
( ) I don't feel qualified to judge about the English language and style

Yes

Can be improved

Must be improved

Not applicable

Does the introduction provide sufficient background and include all relevant references?

(x)

( )

( )

( )

Are all the cited references relevant to the research?

(x)

( )

( )

( )

Is the research design appropriate?

(x)

( )

( )

( )

Are the methods adequately described?

( )

(x)

( )

( )

Are the results clearly presented?

( )

(x)

( )

( )

Are the conclusions supported by the results?

( )

(x)

( )

( )

Comments and Suggestions for Authors

The paper is best described as an empirical report of using the DigitSeis software to perform a digitization of historic seismograms from pre-standardization era. A great introduction is given to the overall problems related to making the historic seismology data available for analysis. I would like to point out that the authors did an amazing job of making the paper understandable for researchers in different fields, as well as describing the subject matter in such a way for readers without seismology background to gauge the problemset and the technical challenges associated with working with historical data in this domain.

The main issue I can see with this paper is a vague formulation of the results, in terms of the list of issues and problems the solution for which could advance the methodology and give information for the researchers in this field, as well as the developers of DigitSeis or the ones working on similar software, to improve the design and the algorihtms involved.

1. It is clear that the usage of the software requires manual intervention (by design of the methodology, as well as stemming from the actual experience of the authors), however it is not clear at all how much time was spent on manual corrections, what were the most time-consuming manual tasks, and what steps could theoretically be taken to minimize this time and to improve the process.

2. The analysis of specific issues with computer vision algorithms which generated problems with pre-processing, digitization, classification, vectorization, etc. - was desribed in the text, however the paper, in my opinion, lacks a clear summary of the specific issues in order of their relevance and severity, which could give an idea to computer vision specialists of what are the main issues with the process and how it could be improved. This is especially clear when the authors describe the Figure 21 - it is not clear at all what the readers should take out from this figure and from its discussion - whether the obtained results are good, bad, acceptable for further analysis, etc.

3. In several parts of the paper authors refer to machine learning tools (e.g. lines 107, 275, 556-558). In the introduction (line 107) the authors state that they are going to identify how ML tools could be used to improve the process (which, in my opinion, the authors do not do), in section 2.5 the authors claim that some specific tasks could only be solved by using ML tools without references or any kind of evidence and corroboration, and, finally, in the conclusion the authors state that the ML tools could be useful for improving the process and produce a vague suggestion that "ML solution of combining DigitSeis with Python" would help somehow. All this is very vague and does not give any insight of what is expected from any kind of ML technologies for improving the process. In order to add this to discussion the authors should clearly state the specific problems which in their opinion could be solved by ML tools, otherwise all these are simply empty statements.

A list of small typos and issues found:

  - DigitSeis software is first mentioned in section 1.4 (line 97), please add the reference citation there, as it is only cited later.

  - line 463 - "revolutions per hours" - probably meant "per hour".

  - line 540 - "very used tool" - probably meant "very useful tool".

  - lines 549-550 - sentence ".. this software limits making the fully automatic analysis impossible" - consider rephrasing, probably meant "... the limits of this software make the fully automatic analysis impossible".

  - line 590 - "available to through open access" - probably meant "available through open access".

Overall, I think that the paper would be interesting to the community, but it would greatly benefit if a clear summary of all found issues in the process were given, with suggestions of what improvements could be made in the software (from the point of the user and from the point of the properties of analyzed data), as well as more clear presentation of the results.

Submission Date

17 October 2022

Date of this review

20 Nov 2022 11:08:04

Reviewer 3 Report

The paper may need several grammatical corrections but overall it's well written. However, my main concerns are with the contribution, novelty and significance of the research. As far as I can tell, the paper is simply using a software that was already designed and developed for the specific task of digitizing analog seismographs. The software DigiSeis was originally introduced in 2014/2015, so has  been around for about 7 years now. This software was developed and made freely available as a part of this project: http://www.seismology.harvard.edu/HRV/archive.html

Most of the paper, it seems, is simply following the steps described in great detail in the actual user manual of the software available here: http://seismology.harvard.edu/downloads/DigitSeis/DigitSeis1.5/DigitSeis_v_1_5_FULLMANUAL.pdf

If this paper was submitted 5-6 years ago it would have made sense as at that time DigitSeis software was not well tested. Now it seems like it already has been used to digitize many seismographs by the original research group as evidenced here: http://www.seismology.harvard.edu/HRV/1952.html . As far as I can tell, scanned images and digitized traces are available for many years worth of seismograms.

In light of the above, I am not sure if there is any new contribution of this research that should be published.

Another aspect of the paper is the claim to introduce ML for fully automated data processing for seismographs.

Lines 555-558: "For the fully automated data processing and rapid building of big data corpus of digitised seismograms, the advanced solutions of ML techniques are useful. As a suggestion for this, we propose a ML solution of combining DigitSeis with Python." Unfortunately, I don't see any evidence of the use of "Machine Learning" to automate any of the tasks which require manual processing where DigitSeis fails to produce accurate results. As far as I can tell, the outputs of the semi-automated process were simply read with Python and some validation is done with Matplotlib. That is not Machine Learning. If some machine learning algorithms have been used or developed to automate this task then please specify clearly.

In it's present form, I think this work could be presented in workshop but unfortunately I don't believe it is fit novel/scientific enough to be published as a technical paper.

Author Response

Dear Editors of Sensors,

Please find attached the revised version of the paper. We have carefully followed the comments and suggestions of the reviewers and corrected the manuscript accordingly.

All the corrections in the text are marked up yellow for Track Changes.

The replies to the comments of the reviewers are listed below.

Using the opportunity, we thank the reviewers for careful reading of the paper which improved the initial version of the manuscript.

With kind regards, - Authors.

04.12.2022

Reviewer 3

No

Reviewer’s Comments

Authors’ actions

1

Moderate English changes required

The manuscript is proofread throughout. We have corrected all occasional typesetting misprints and minor grammar mistakes (spelling, punctuation) where necessary. Grammar errors are corrected and misprints are checked everywhere in the text.

2

The paper may need several grammatical corrections but overall it's well written. However, my main concerns are with the contribution, novelty and significance of the research. As far as I can tell, the paper is simply using a software that was already designed and developed for the specific task of digitizing analog seismographs. The software DigiSeis was originally introduced in 2014/2015, so has  been around for about 7 years now. This software was developed and made freely available as a part of this project: http://www.seismology.harvard.edu/HRV/archive.html

We have elaborated on explaining the contribution, novelty and significance of the research, which are described in the Introduction. We restructured this section and divided it into the four parts: 1.1. Background (here we briefly describe the phenomenon of the seismicity of the Earth, associated geophysical processes, mention the development and progress of the seismogram recordings); subsection 1.2. Motivation (here we raise the question of the importance of seismograms, in which domains they are applicable, and why the methods of accurate processing of seismogram are essential for data interpretation); 1.3. Related Work (here we describe the existing methods of seismogram processing and present many examples of data vectorization, explain briefly some key issues of digitizing in terms of algorithms and data processing); subsection 1.4. Contribution (in this subsection we outlines our contribution to the seismogram processing, described the goals and objectives of this work, presented methods and scope of this research, and mentioned the large volume of data that should be digitized rapidly using effective and robust methods). We mentioned the advantages and benefits of the proposed frameworks and described the workflow.

3

Most of the paper, it seems, is simply following the steps described in great detail in the actual user manual of the software available here: http://seismology.harvard.edu/downloads/DigitSeis/DigitSeis1.5/DigitSeis_v_1_5_FULLMANUAL.pdf If this paper was submitted 5-6 years ago it would have made sense as at that time DigitSeis software was not well tested. Now it seems like it already has been used to digitize many seismographs by the original research group as evidenced here: http://www.seismology.harvard.edu/HRV/1952.html As far as I can tell, scanned images and digitized traces are available for many years worth of seismograms. In light of the above, I am not sure if there is any new contribution of this research that should be published.

We have added more explanations regarding the dataset recorded in Royal Observatory of Belgium (ROB) and the importance of the vectorising the archived dataset for retrospective data modelling, analysis and archive maintenance. This work is a part of the SeismoStorm project which aims at processing the archives of the dataset containing historical recordings from Belgian seismic station. To the best of our knowledge, there are analogues of this work, since ROB is the only main seismological station of Belgium. The scope and task of this work is to vectorise a large archive of data originated from the Belgian seismological stations, which is important for regional seismic monitoring in Belgium. To this end. We used the DigitSeis as a processing tool and an effective instrument of data processing and vectorising raster images into the digital format.

4

Another aspect of the paper is the claim to introduce ML for fully automated data processing for seismographs. Lines 555-558: "For the fully automated data processing and rapid building of big data corpus of digitised seismograms, the advanced solutions of ML techniques are useful. As a suggestion for this, we propose a ML solution of combining DigitSeis with Python." Unfortunately, I don't see any evidence of the use of "Machine Learning" to automate any of the tasks which require manual processing where DigitSeis fails to produce accurate results. As far as I can tell, the outputs of the semi-automated process were simply read with Python and some validation is done with Matplotlib. That is not Machine Learning. If some machine learning algorithms have been used or developed to automate this task then please specify clearly. In it's present form, I think this work could be presented in workshop but unfortunately I don't believe it is fit novel/scientific enough to be published as a technical paper.

This paragraph is updated. The phrases regarding the ML are moved to the Discussion where we discuss future work and recommended further steps of this research. Indeed, the DigitSeis does not include the ML components, and the data were read into Python and with validation is done using Matplotlib for post-processing where the data were imported using the using .mat format (Matlab). However, the integration of data was possible using export/import of data and we performed it as a separate step. The recommendations to include a smooth integration of data are added in Discussion section. Since these steps of data processing are now separated, the data processing in the future work should include more ML aspects. We use the machine learning algorithms using Python as a continue of this work, developed to automate this task, as briefly mentioned in the Discussion. This is a continuation of this work, necessary for processing large amount of data, which will be described in the following paper continuing current study.

Original Open Review

English language and style

( ) English very difficult to understand/incomprehensible
( ) Extensive editing of English language and style required
(x) Moderate English changes required
( ) English language and style are fine/minor spell check required
( ) I don't feel qualified to judge about the English language and style

Yes

Can be improved

Must be improved

Not applicable

Does the introduction provide sufficient background and include all relevant references?

(x)

( )

( )

( )

Are all the cited references relevant to the research?

(x)

( )

( )

( )

Is the research design appropriate?

(x)

( )

( )

( )

Are the methods adequately described?

(x)

( )

( )

( )

Are the results clearly presented?

(x)

( )

( )

( )

Are the conclusions supported by the results?

(x)

( )

( )

( )

Comments and Suggestions for Authors

The paper may need several grammatical corrections but overall it's well written. However, my main concerns are with the contribution, novelty and significance of the research. As far as I can tell, the paper is simply using a software that was already designed and developed for the specific task of digitizing analog seismographs. The software DigiSeis was originally introduced in 2014/2015, so has  been around for about 7 years now. This software was developed and made freely available as a part of this project: http://www.seismology.harvard.edu/HRV/archive.html

Most of the paper, it seems, is simply following the steps described in great detail in the actual user manual of the software available here: http://seismology.harvard.edu/downloads/DigitSeis/DigitSeis1.5/DigitSeis_v_1_5_FULLMANUAL.pdf If this paper was submitted 5-6 years ago it would have made sense as at that time DigitSeis software was not well tested. Now it seems like it already has been used to digitize many seismographs by the original research group as evidenced here: http://www.seismology.harvard.edu/HRV/1952.html . As far as I can tell, scanned images and digitized traces are available for many years worth of seismograms.

In light of the above, I am not sure if there is any new contribution of this research that should be published.

Another aspect of the paper is the claim to introduce ML for fully automated data processing for seismographs.

Lines 555-558: "For the fully automated data processing and rapid building of big data corpus of digitised seismograms, the advanced solutions of ML techniques are useful. As a suggestion for this, we propose a ML solution of combining DigitSeis with Python." Unfortunately, I don't see any evidence of the use of "Machine Learning" to automate any of the tasks which require manual processing where DigitSeis fails to produce accurate results. As far as I can tell, the outputs of the semi-automated process were simply read with Python and some validation is done with Matplotlib. That is not Machine Learning. If some machine learning algorithms have been used or developed to automate this task then please specify clearly.

In it's present form, I think this work could be presented in workshop but unfortunately I don't believe it is fit novel/scientific enough to be published as a technical paper.

Submission Date

17 October 2022

Date of this review

24 Nov 2022 03:55:16

Reviewer 4 Report

The article focuses on how to automatically datatype the recorded old data to convert it from raster format to vector format. And the DigitSeis method used in this study can perform a complete workflow including pattern recognition, classification, digitization, correction and conversion of tiff to digital output. This study contributes to the signal processing methods for archiving seismograms. It can be published after minor revision. Followed concerns should be noted.

1 Authors need to provide the reason why 82 in Section 2.5.3 is the most appropriate image threshold for the selected sample.
    2 Please elaborate on how the threshold parameters are set for seismic record classification in Figure 9DigitSeis.
    3 There is a typographical formatting problem in lines 345 and 346.
    4 Authors should make explanation for the basis of setting the reference time point in Section 3.3.
    5 Typographical formatting in rows 345 and 346 needs to be refined.

Author Response

Dear Editors of Sensors,

Please find attached the revised version of the paper. We have carefully followed the comments and suggestions of the reviewers and corrected the manuscript accordingly.

All the corrections in the text are marked up yellow for Track Changes.

The replies to the comments of the reviewers are listed below.

Using the opportunity, we thank the reviewers for careful reading of the paper which improved the initial version of the manuscript.

With kind regards, - Authors.

04.12.2022.

Reviewer 4

No

Reviewer’s Comments

Authors’ actions

1

Are the methods adequately described? – Can be improved.

The Methodology section is improved and updated. Also, we have proofread the manuscript throughout, corrected all the occasional typesetting misprints and minor grammar mistakes where required (spelling, punctuation). The grammar errors and misprints are checked in the text.

2

The article focuses on how to automatically datatype the recorded old data to convert it from raster format to vector format. And the DigitSeis method used in this study can perform a complete workflow including pattern recognition, classification, digitization, correction and conversion of tiff to digital output. This study contributes to the signal processing methods for archiving seismograms. It can be published after minor revision. Followed concerns should be noted.

Many thanks for the commenting and supporting of our manuscript. Yes, we processed a dataset in raster format and converted them to vector, using the DigitSeis techniques. The workflow will be used to continue our research (and other similar works) in the future, to process with historical archives of seismograms. The workflow of DigitSeis is promising and includes many steps of data processing which we demonstrated in our paper. We improved the manuscript according to your (and other reviewers’) comments and marked up the changes in yellow color for Track Changes.

3

1 Authors need to provide the reason why 82 in Section 2.5.3 is the most appropriate image threshold for the selected sample.

Added the following explanation in the text: “The image threshold of minimized the number of misclassified pixels compared to other values, that is, the distributions of the gray level values in pixels is best distinguished by this values, which makes up the object of seismic lines well separated from the background of seismograph. After we compared other values of image thresholds, we noted that the level of 82 well separates the lines of seismic signals (as foreground pixels) from the scanned paper (as background pixels). Thus, the empirical selection of this value improved the results of image processing.”

4

2 Please elaborate on how the threshold parameters are set for seismic record classification in Figure 9 DigitSeis.

Added following explanation in the text: “The threshold parameters for seismogram classification in DigitSeis include several values, as follows. The image threshold is set to 82, as explained above. The time mark width indicates the distance between the tiny marks representing minutes. Since they vary in various seismograms due to the paper recording of each seismograph, the width is detected manually on the screen, and the lowest value is selected, to ensure that we do not skip the possible cases. The time mark offset is an option useful for the seismograms where traces are interrupted each minute by an offset time mark. In our case, we did not have time marks; instead the minutes are indicated by gaps regularly repeating as tiny pauses in recording each minute. Object thickness represent the natural width of line recorded by drum, visible on the paper and detectable by the computer vision algorithm. In this case, we had the value of 25 pixels.”

5

3 There is a typographical formatting problem in lines 345 and 346.

Corrected in both issues.

6

4 Authors should make explanation for the basis of setting the reference time point in Section 3.3.

Explained in the following sentence: “The basis of setting the reference time point is according to the metadata presented in each seismogram. When originally recorded, the analog seismograms included the information regarding the date, hour and minutes of the start of the seismic recording. This original information of time series data from the digital records of Uccle station was copied and used to set the reference time.”

7

5 Typographical formatting in rows 345 and 346 needs to be refined.

Corrected (in phrase “The algorithm of classification identified lines and traces, and discriminated them from noise and time gaps”).

Original Report Reviewer-4

Open Review

English language and style

( ) English very difficult to understand/incomprehensible
( ) Extensive editing of English language and style required
( ) Moderate English changes required
(x) English language and style are fine/minor spell check required
( ) I don't feel qualified to judge about the English language and style

Yes

Can be improved

Must be improved

Not applicable

Does the introduction provide sufficient background and include all relevant references?

(x)

( )

( )

( )

Are all the cited references relevant to the research?

(x)

( )

( )

( )

Is the research design appropriate?

(x)

( )

( )

( )

Are the methods adequately described?

( )

(x)

( )

( )

Are the results clearly presented?

(x)

( )

( )

( )

Are the conclusions supported by the results?

(x)

( )

( )

( )

Comments and Suggestions for Authors

The article focuses on how to automatically datatype the recorded old data to convert it from raster format to vector format. And the DigitSeis method used in this study can perform a complete workflow including pattern recognition, classification, digitization, correction and conversion of tiff to digital output. This study contributes to the signal processing methods for archiving seismograms. It can be published after minor revision. Followed concerns should be noted.

1 Authors need to provide the reason why 82 in Section 2.5.3 is the most appropriate image threshold for the selected sample.
2 Please elaborate on how the threshold parameters are set for seismic record classification in Figure 9DigitSeis.
3 There is a typographical formatting problem in lines 345 and 346.
4 Authors should make explanation for the basis of setting the reference time point in Section 3.3.
5 Typographical formatting in rows 345 and 346 needs to be refined.

Submission Date

17 October 2022

Date of this review

25 Nov 2022 11:14:25
